# MMP14 cleaves PTH1R in the chondrocyte-derived osteoblast lineage, curbing signaling intensity for proper bone anabolism

**Tsz Long Chu[1,2], Peikai Chen[1,3], Anna Xiaodan Yu[1], Mingpeng Kong[1], Zhijia Tan[1,3], Kwok Yeung Tsang[1], Zhongjun Zhou[1], Kathryn Song Eng Cheah[1]\***

[1]School of Biomedical Sciences, University of Hong Kong, Pokfulam, Hong Kong SAR, Hong Kong; [2]Department of Physiology and Pharmacology, Karolinska Institutet, Solna, Sweden; [3]Department of Orthopaedics and Traumatology, The University of Hong Kong-Shenzhen Hospital, Shenzhen, China

**Abstract** Bone homeostasis is regulated by hormones such as parathyroid hormone (PTH). While PTH can stimulate osteo-progenitor expansion and bone synthesis, how the PTH-signaling intensity in progenitors is controlled is unclear. Endochondral bone osteoblasts arise from perichondrium-derived osteoprogenitors and hypertrophic chondrocytes (HC). We found, via single-cell transcriptomics, that HC-descendent cells activate membrane-type 1 metalloproteinase 14 (MMP14) and the PTH pathway as they transition to osteoblasts in neonatal and adult mice. Unlike *Mmp14* global knockouts, postnatal day 10 (p10) HC lineage-specific *Mmp14* null mutants (Mmp14ΔHC) produce more bone. Mechanistically, MMP14 cleaves the extracellular domain of PTH1R, dampening PTH signaling, and consistent with the implied regulatory role, in Mmp14ΔHC mutants, PTH signaling is enhanced. We found that HC-derived osteoblasts contribute ~50% of osteogenesis promoted by treatment with PTH 1–34, and this response was amplified in Mmp14ΔHC. MMP14 control of PTH signaling likely applies also to both HC- and non-HC-derived osteoblasts because their transcriptomes are highly similar. Our study identifies a novel paradigm of MMP14 activity-mediated modulation of PTH signaling in the osteoblast lineage, contributing new insights into bone metabolism with therapeutic significance for bone-wasting diseases.

## Editor's evaluation

The authors present novel findings that PTH signaling plays a significant role in bone formation in hypertrophic chondrocyte (HC)-derived osteoblasts and MMP14 cleaves PTH1R and inhibits PTH signaling. These studies significantly contribute to our understanding of molecular mechanisms of postnatal bone formation and adult bone remodeling, especially the HC cells in this process. The study was well-designed and well-conducted. The data in this study are convincing and support the conclusion made by the authors.

## Introduction

Developing and maintaining appropriate proportions of cells of the osteoblast lineage and other cell types (e.g., osteoclasts, endothelial cells, adipocytes, stem cells) in the bone marrow are crucial for healthy bones. These cell types together maintain bone mass via concerted activities of bone building (anabolic) by osteoblasts versus bone resorption (catabolic) by osteoclasts, with input from

**\*For correspondence:**
kathycheah@hku.hk

factors produced by osteocytes to orchestrate the remodeling process (*Florencio-Silva et al., 2015*). Through this continuous remodeling, and balanced anabolic and catabolic activity, a homeostatic condition is achieved (*Karsenty et al., 2009*). This homeostasis is regulated by growth factors and cytokines produced within bone and systemic factors such as parathyroid hormone (PTH) and estrogens (*Wein and Kronenberg, 2018*; *DiGirolamo et al., 2012*). An imbalance in proportions of these cells in bone impairs bone resorption and formation, which can result in diseases of bone mass such as osteopenia, osteoporosis, and osteopetrosis (*Ayturk et al., 2020*; *Kenkre and Bassett, 2018*).

Long bones form by endochondral ossification, a process in which chondrocytes differentiate, proliferate, mature, and become hypertrophic, forming a cartilaginous growth plate that coordinates longitudinal bone growth and is recapitulated in fracture repair (*Hu et al., 2017*; *Javaheri et al., 2018*). Hypertrophic cartilage is remodeled by matrix metalloproteinases (MMPs) (reviewed in *Tsang and Cheah, 2019*). The chondrocyte differentiation cascade is coordinated by many transcription factors and signaling pathways, including reciprocal signaling via the parathyroid hormone-related protein (PTHRP)-Indian hedgehog (IHH) feedback loop (*Kronenberg, 2003*; *Long and Ornitz, 2013*). Resting chondrocytes secrete PTHRP, which binds to parathyroid hormone 1 receptor (PTH1R) expressed by pre-hypertrophic chondrocytes, acting to delay their differentiation to hypertrophic chondrocytes (HCs) (*Mizuhashi et al., 2018*; *Newton et al., 2019*). HCs specifically produce collagen type X (*Col10a1*), downregulate PTH1R, and undergo distinct phases of cell enlargement for bone elongation (*Cooper et al., 2013*; *Yang et al., 2014a*). Recent work has shown that osteoblasts in trabecular bone are derived from HCs in the growth plate and from osteoprogenitors in the perichondrium that accompany invading blood vessels (*Yang et al., 2014a*; *Maes et al., 2010*; *Yang et al., 2014b*). HCs have been shown to contribute to the full spectrum of cells in the osteoblast lineage in endochondral bone, trabeculae, endosteal/endocortical bone, including osteocytes, and, to a minor degree, bone marrow stromal cells and adipocytes (*Yang et al., 2014a*; *Yang et al., 2014b*; *Park et al., 2015*; *Tan et al., 2020*). HC transformation to osteoblasts also occurs in bone healing (*Hu et al., 2017*; *Yang et al., 2014a*; *Yang et al., 2014b*). The functional importance of the HC lineage has been demonstrated in mice, which have reduced bone mass as a consequence of ablating β-catenin and *Irx3/5* genes specifically in HCs (*Tan et al., 2020*; *Houben et al., 2016*). However, it is not known whether the HC-derived osteogenic lineage is molecularly distinct from the non-HC lineage (contributed by the perichondrium, periosteum, bone marrow mesenchymal stem cells), which specific MMPs act at the chondro-osseous junction to facilitate the transition of HCs to osteoblasts, and what physiological contribution(s) they make to maintain bone homeostasis and anabolism in response to extrinsic signals.

Here, using single-cell transcriptomics, we characterized the HC-derived osteoblast populations in endochondral bone and found that their transcriptomes are broadly similar to that of non-HC-derived osteoblasts. We found that HC-derived osteoblasts activate membrane-type 1 metalloproteinase 14 (MMP14) and the PTH/PTH1R pathway upon transition into sub-chondral bone. Using mouse mutants and biochemical approaches, we identified the cleavage of PTH1R by MMP14 in HC-derived osteoblasts as a novel mechanism that curbs the intensity of PTH signaling. Like their non–HC-derived counterparts, HC-derived osteoblasts respond to exogenous PTH treatment and contribute significantly to the bone anabolic response. MMP14 modulation of PTH signaling intensity controls the differentiation and survival of osteoblasts and thereby the anabolic response to PTH in building bone mass.

## Results

### HC derivatives activate MMP14 as they transition to become osteoblasts

To reveal key transcriptional changes and MMPs that could potentially facilitate the translocation of HC lineage cells as they cross the chondro-osseus junction to contribute to forming trabecular bones, we exploited *Col10a1*$^{Cre/+}$ (C10Cre); *Rosa26*$^{LSL-tdTomato/LSL-tdTomato}$ (RtdT) (abbreviated C10Cre;RtdT) mice wherein the HC-derived osteoblast lineage is marked by the expression of tdTomato because of the specific activity of Cre recombinase on the RtdT *Cre* reporter in HCs (*Yang et al., 2014a*; *Tan et al., 2020*; *Madisen et al., 2010*). We performed single-cell transcriptomics (scRNAseq) on the tibiae of C10Cre;*Irx3*$^{+/\Delta HC}$*Irx5*$^{+/-}$; RtdT $^{td/+}$ mice at postnatal day 6 (P6) (phenotypically normal) (*Tan et al., 2020*)

and on that of C10Cre;*Mmp14*^+/F^; RtdT; Col1a1-GFP (C1-GFP) mice (also phenotypically normal) at P56 (*Figure 1A*; 'Materials and methods').

Clustering analyses of the P6 tibia scRNAseq data, identified 12 populations, which, from their molecular signatures, included eight chondro-osteogenic clusters and four other cell types (*Figure 1B* and 'Materials and methods'). By calculating the cluster signatures and referencing known markers in the literature (*Tan et al., 2018*; *Tikhonova et al., 2019*; *Wai et al., 1998*), we annotated the five chondrogenic clusters as resting (e.g., *Pthlh*), proliferating (e.g., *Top2a, Mki67*), maturing (e.g., *Col9a2, Matn1*), pre-hypertrophic (pre-HCs) (e.g., *Ihh, Fgfr3, Pth1r*), and hypertrophic chondrocytes (*Col10a1*) (*Figure 1B*, *Figure 1—figure supplement 1A*, *Figure 1—source data 1*). Similarly, the three osteogenic clusters were identified as immature (*Mmp14, Pdgfrb*) and mature (*Ifitm5, Bglap*) osteoblasts, and osteocytes (corresponding to terminally differentiated osteoblasts, *Sost1, Dmp1, Phex*) (*Figure 1B and C*, *Figure 1—figure supplement 1A*, and 'Materials and methods').

We identified and distinguished HC-derived osteoblasts from non HC-derived osteoblasts by considering the expression of tdTomato transcripts, as defined by the detection of sequences encoding tdTomato and a surrogate sequence (WPRE) (*Higashimoto et al., 2007*; *Loeb et al., 1999*; 'Materials and methods,' *Figure 1—figure supplement 1A, E–G*). Gene Ontology (GO) enrichment analyses suggest a series of developmental events consistent with the onset of endochondral ossification (*Figure 1—figure supplement 1B–D*, *Figure 1—source data 2*; *Maes et al., 2010*) and confirm the lineage continuum of HCs and osteoblasts. Immature osteoblasts represent an important early stage in the HC to osteoblast transition and 18.1% of them were estimated to be HC-derived (tdTomato expressing) (*Figure 1C*, *Figure 1—figure supplement 1H*).

Given the need for HCs to be released from the cartilage matrix as they move across the chondro-osseous junction, MMP genes are candidates, especially those expressed at that region and in the adjacent sub-chondral bone. Amongst the *Mmp's, Mmp13* was expressed in both the HC and immature Ob populations (*Figure 1C*, *Figure 1—source data 1* and *Figure 1—source data 3*; *Yang et al., 2014a*; *Tan et al., 2018*), consistent with the literature (*Stickens et al., 2004*). By contrast,*Mmp14*, which encodes a membrane-anchored MMP (*Zhou et al., 2000*), was not expressed in HCs but was enriched in immature osteoblasts (*Figure 1C*, *Figure 1—figure supplement 1A*, *Figure 1—source data 1*). Consistent with the scRNAseq results, immunostaining for MMP14 detected no protein in HCs and strong expression in LacZ-labeled HC-descendant cells, especially those located at the chondro-osseous junction (*Figure 1D*) (identified by co-staining with LacZ in C10Cre; *Rosa26*^LSL-LacZ/LSL-LacZ^ [RLacZ] mice; *Yang et al., 2014a*).

We next investigated whether the molecular characteristics of the osteogenic signatures in HC-derived osteoblasts persisted in more mature mice at P56 (*Figure 1E–J*, *Figure 1—figure supplement 1I and J*, *Figure 1—source data 3*). We integrated the chondro-osteogenic cell clusters in the P6 (*Figure 1E*; 3163 cells) and P56 (*Figure 1F*; 430 cells) samples, whereby two major clusters corresponding to chondrogenic and osteogenic cells were identified (*Figure 1G*). We focused on the 1256 osteogenic cells of P6 and P56 combined, wherein four subpopulations were identified, including osteo-progenitors (marked by *Grem1, Vcam1, Lpl, Steap4*, and *Col4a1/2*), immature osteoblasts (*Postn, Slc20a2*), proliferating osteoblasts (*Top2a, Mki67*, and *Birc5*), and mature osteoblasts (*Bglap, Phex, Dmp1, Col22a1, Ifitm5*, and *Smpd3*) (*Figure 1H and L*, *Figure 1—figure supplement 1I*, *Figure 1—source data 3*). Interestingly, although both P6 (6.8%, or 215 out of 3163 cells) and P56 (9.8%, or 42 out of 408 cells) retained certain proportions of proliferating chondrocytes (*Figure 1G*), the P56 had much fewer proliferating osteoblasts (only 1 cell, or 0.2%) than the P6 (190 cells, or 6%) (*Figure 1I*, *Figure 1—source data 3*), which probably reflects maturation of the osteoblast lineage by P56. The other three populations varied between 20% and 40% in both time points (*Figure 1I*). Using the tdTomato expression criterion, we found that the HC-derived osteoblasts appeared randomly dispersed in the combined osteogenic cells of P6 and P56 (*Figure 1J*). The percentages of HC-derived osteoblasts range between 17.4 and 35%, except the proliferating osteoblasts of P56, which had only one cell and it was not HC-derived (*Figure 1K*). Overall, 23.8% of the P6 and 31.6% of the P56 osteogenic cells were HC-derived. We cannot exclude that the variations in relative percentages of cell populations detected at P6 and P56 were related to the limits in sensitivity in detecting tdTomato mRNAs and differences in the ease of releasing deeply embedded cells within the bone matrix versus more superficially located ones. Nonetheless, these frequencies are in agreement with the broad ranges (18–60%) reported previously for HC-derived

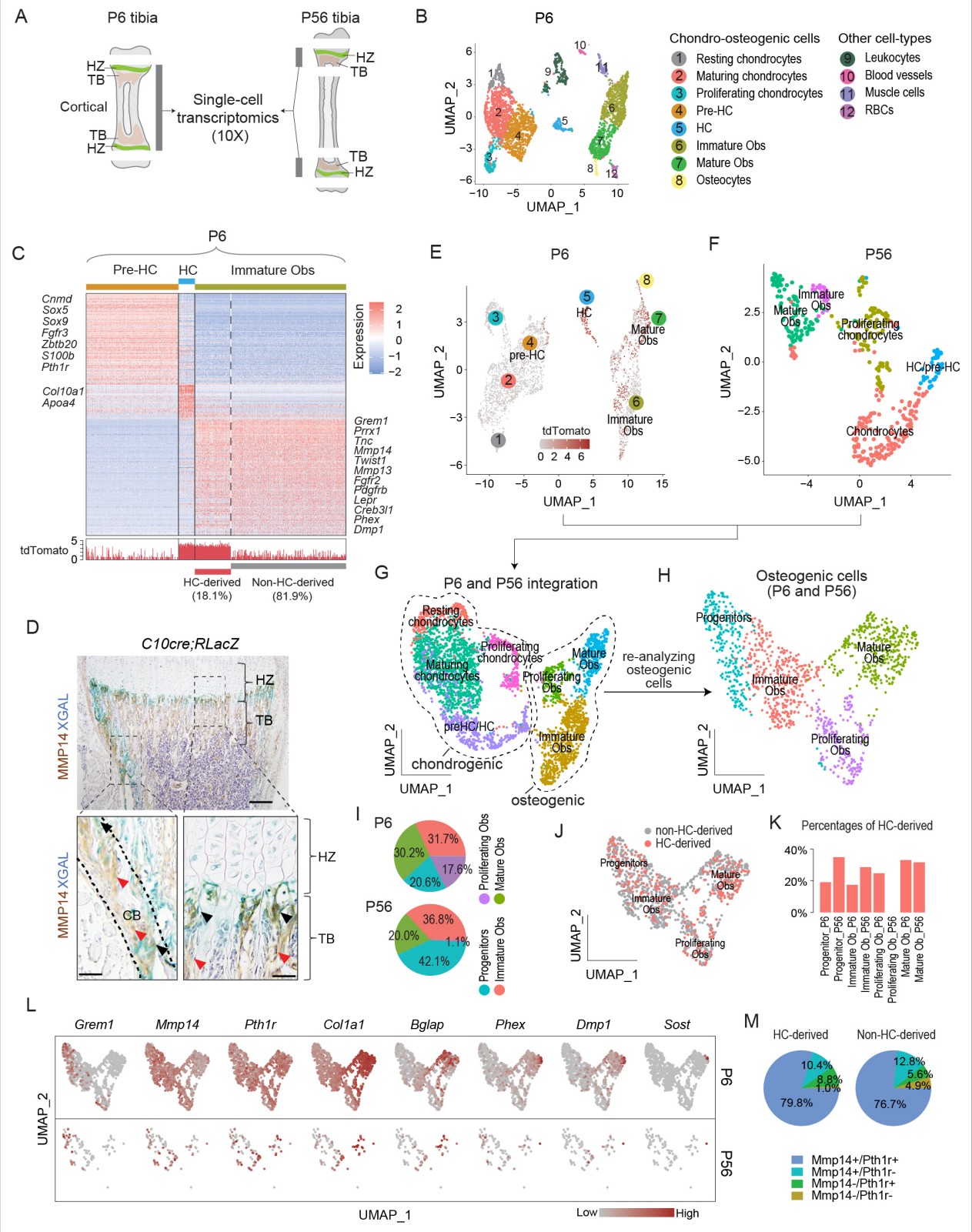

**Figure 1.** Single-cell transcriptomic analyses of endochondral bone sub-populations at P6 and P56. (**A**) Schematic diagram showing the isolation of osteochondrogenic cells from mouse tibia for single-cell transcriptomics (scRNAseq). (**B**) Scatter plot showing the 3420 single cells, 3163 of which belong to the chondro-osteogenic clusters, as identified in the P6 sample by dimension reduction method UMAP. HC: hypertrophic chondrocytes; Obs: osteoblasts; RBC: red blood cells. (**C**) Upper panel: heatmap showing the expression patterns of P6 signature genes in the preHC, HC, and immature Ob

*Figure 1 continued on next page*

*Figure 1 continued*

clusters, when comparing in a one-vs.-all manner among the three. Representative signature genes of interest are listed by the side. Lower panel: signal track indicating the level of tdTomato expression inferred from the WPRE signal. (**D**) Representative immunostaining showing the spatial distribution of MMP14 and X-gal stained (blue) in hematoxylin-stained sections from C10Cre;RLacZ tibia. HZ, hypertrophic zone; TB, trabecular region; CB, cortical region. Black arrows in enlarged figures indicate LacZ+;MMP14+ cells. Red arrows indicate Mmp14+;LacZ- cells. Dotted line draws chondro-osseous junction. Scale bar 200 µm, 50 µm (magnified picture). (**E–G**) Scatter plots showing the integration (**G**) of the 3163 and 430 chondro-osteogenic cells of the P6 (**E**) and the P56 (**F**) mice, respectively. (**H**) Scatter plot showing the reanalyzed UMAP of the 1219 cells osteogenic cells (1124 from P6 and 95 from P56). (**I**) Piecharts showing the percentages of osteoblast subpopulations in both time points. (**J**) Scatter plot showing the distribution of HC-derived Obs in the 1219 Obs in (**H**). (**K**) Bar chart showing the percentages of HC-derived Obs in the Ob subpopulations of both time points. (**L**) A panel of scatter-plots showing the expression pattern of a list of selected markers, representing the different osteoblast differentiation states. (**M**) Piechart showing the percentages of *Mmp14* and *Pth1r* co-expression in HC- and non-HC-derived Obs.

The online version of this article includes the following source data and figure supplement(s) for figure 1:

**Source data 1.** Cluster signatures for the sub-populations in the P6 sample (*Figure 1E*).

**Source data 2.** Enriched Gene Ontology terms for the subpopulations at P6.

**Source data 3.** Cluster signatures for the sub-populations at P56 (*Figure 1F*), and for the sub-populations of P6 and P56 combined (*Figure 1J* and *Figure 1—figure supplement 1I*).

**Source data 4.** Differentially expressed genes (DEGs) between hypertrophic chondrocyte (HC)- and non-HC-derived Obs in the combined osteogenic cells of P6 and P56.

**Figure supplement 1.** Integrative analyses of endochondral bone subpopulations at P6 and P56.

osteoblasts from a range of developmental and postnatal ages (*Yang et al., 2014a*; *Yang et al., 2014b*; *Long et al., 2022*).

We also assessed how similar HC-derived and non-HC-derived cells are, and detected the 38 genes (*Figure 1—source data 4*) that were differentially expressed between the two lineages, with 10 genes expressed higher in the HC-derived and 28 genes in the non-HC-derived (*Figure 1—figure supplement 1J*). The genes higher in the HC-derived include some mature osteoblast markers, such as *Ibsp*, and *Car3*; and the genes higher in the non-HC-derived include such as *Postn*, *Igfbp5*, *Col3al* (*Figure 1—figure supplement 1J*, *Figure 1—source data 4*). This small set of differentially expressed genes may reflect heterogeneity and differences in maturities between osteoblasts of different origins, the significance of which will be investigated in future studies. Overall, there was a broad similarity in transcriptomic characteristics for the HC-derived and non-HC-derived osteoblasts.

## Global MMP14 is required for proper translocation of HC derivatives to trabecular bone

*Mmp13* and *Mmp14* are candidate facilitators for the translocation of HCs to the subchondral space. *Mmp13* is expressed in late HCs (*Yang et al., 2014a*; *Tan et al., 2018*) as well as osteoblasts. It was also expressed in the P6 chondrocyte populations (*Figure 1—source data 1*) and immature osteoblasts at P56 (*Figure 1—source data 3*). Global *Mmp13* knockout mice display abnormal growth plates with disrupted terminal hypertrophy and increased trabecular bone which could reflect an impact on the HC-Ob lineage continuum. However, while *Col1a1*-cre conditional (acting on osteoblasts) *Mmp13* knockout mice showed increased trabecular bone, *Col2a1*-cre *Mmp13* conditional mutants (acting on chondrocytes and perichondrium) did not, but displayed trabecular bone irregularities that were attributed to remodeling defects (*Stickens et al., 2004*). By contrast, MMP14 is expressed specifically at the chondro-osseous junction in immature HC-derived osteoblasts but not in HCs themselves (*Figure 1C and D*). Amongst MMPs, *Mmp14* null mutants display the most severe skeletal phenotypes, including loss of trabecular bone, over-activity of osteoclasts, impaired angiogenesis, and reduced calvarial ossification (*Zhou et al., 2000*; *Holmbeck et al., 1999*; *Hikita et al., 2006*; *Chan et al., 2012*), consistent with expression of *Mmp14* by many cell types, such as skeletal progenitors, osteoblasts, osteoclasts, endothelial cells, and bone marrow stromal cells (*Figure 2—figure supplement 1A*; *Chan et al., 2012*; *Jin et al., 2011*; *Gonzalo et al., 2010*; *Langlois et al., 2004*; *Attur et al., 2020*). The impaired bone formation in *Mmp14*-deficient mice and the expression of MMP14 in HC-derived osteoblasts raised the possibility that MMP14 regulates their differentiation. Interestingly, *Mmp14* is co-expressed with *Pth1r*, a key regulator of chondrocyte hypertrophy and osteogenesis (*Calvi et al., 2001*; *Hirai et al., 2011*; *Lanske et al., 1999*), where >75% of both

HC-derived and non-HC-derived lineages co-expressed *Mmp14+/Pth1r+* (*Figure 1L and M*). The impact of chondrocyte-specific knockout of *Mmp14* on trabecular bone formation is not known. Given MMP14's pivotal role in trabecular bone formation and its co-expression with key regulators of osteogenesis, we focused on *Mmp14* and asked whether dysfunction in lineage progression of HC-derived cells was responsible for the severe bone deficit in *Mmp14* knockout mice.

We used the C10Cre; *Rosa26*[LSL-YFP/LSL-YFP](RYFP) reporter to lineage trace HC derivatives in global *Mmp14* knockout mice and found an increased accumulation of HC descendants at the chondro-osseous junction compared to WT controls (*Figure 2—figure supplement 2B and C*). There was also a small increase in proliferating HC-derivatives (marked by 5-ethynyl-2'-deoxyuridine [Edu]) in the region immediately below the chondro-osseous junction (*Figure 2—figure supplement 2C*). However, there were fewer proliferating HC-derivatives in further distal regions below the chondro-osseous junction, consistent with their accumulation there and impaired translocation into the trabecular bone (*Figure 2—figure supplement 2C*).

## Intrinsic MMP14 activity in HC derivatives is not essential for their translocation to trabecular bone

To investigate whether the stagnation of HC descendants under the chondro-osseous junction reflected impaired translocation to trabecular bone in *Mmp14* global knockouts and was intrinsic to HC-derived cells themselves or involved extrinsic influences from other cell types, we genetically inactivated *Mmp14* in HC-descendants by generating HC-specific conditional *Mmp14*[Flox] (*Mmp14*[F/F]) and *Mmp14*[F/-] mutants using C10Cre (abbreviated as Mmp14ΔHC) (*Figure 2A*). We found that removing *Mmp14* in HC-descendants did not recapitulate the accumulation of HC descendants at the chondro-osseous junction observed in *Mmp14*[-/-] mice (*Figure 2—figure supplement 2C*). We used tamoxifen-mediated pulse-labeling and chasing of *Mmp14*-deficient HC descendants using C10Cre-ERT;*Mmp14*[F/F]; Rtdt to assay the dynamics of their translocation into the subchondral space. We found that the localization and distribution of HC-descendants were unaffected in Mmp14ΔHC mice (*Figure 2B*). These results suggest the aberrant stagnation of HC derivatives at the chondro-osseous junction was not intrinsic to a defect in these cells themselves but might be a consequence of MMP14 deficiency elsewhere, for example, in the invading vascular cells and/or the accompanying osteoprogenitors from the perichondrium that co-migrate with blood vessels to populate the primary spongiosa (*Maes et al., 2010*; *Kusumbe et al., 2014*). Vascular invasion is required for bone formation (*Kusumbe et al., 2014*; *Peng et al., 2020*). We therefore examined the vascular capillaries in *Mmp14*[-/-] mice and found that both vascular density and endothelial cell count (measured using the marker ENDOMUCIN) were decreased compared with control mice (*Figure 2—figure supplement 2D*). In vitro and in vivo studies have shown that *Mmp14* is important for angiogenesis, neovessel formation and migration (*Zhou et al., 2000*; *Chun et al., 2004*). The abnormal vascularization in *Mmp14* null mice may therefore be a major contributor to the compromised translocation of HC-derived cells to the trabecular bone.

## Increased osteogenesis and number of HC descendants in Mmp14ΔHC mutants

To delineate the intrinsic role of *Mmp14* in the HC-lineage and on trabecular bone formation, we characterized the bone phenotype of P10 Mmp14ΔHC mice. Interestingly, microCT analysis revealed a 16% increase in BMD and a 10% increase in trabecular BV/TV in Mmp14ΔHC mice (*Figure 2C*). This increased bone mass in Mmp14ΔHC mutants, in contrast to severe bone deficit in *Mmp14*[-/-], led us to ask whether the increased BMD was a consequence of increased osteogenesis. To elucidate whether increased trabecular bone density and volume in Mmp14ΔHC mice is directly attributable to osteogenic HC descendants, we addressed the fate of HC descendants at postnatal stages (*Figure 2D–F*). Immunofluorescence staining of MMP14 and RFP shows the number of HC descendants was increased in Mmp14ΔHC mice at P10 (*Figure 2D*). This observation could be due to a decrease of TUNEL+ cells at the chondro-osseous junction (*Figure 2E*). By using C1-GFP transgene to co-label osteoblasts, we observed an increased number of HC-derived osteoblasts in Mmp14ΔHC mutants, compared to similar GFP+ RFP cell counts between mutants and controls (*Figure 2F*). The data collectively suggests targeting *Mmp14* in HCs promoted osteogenesis. Consistent with the above findings, the expression domains of osteoblast markers *Col1a1* and *Mmp13* were expanded in Mmp14ΔHC mice

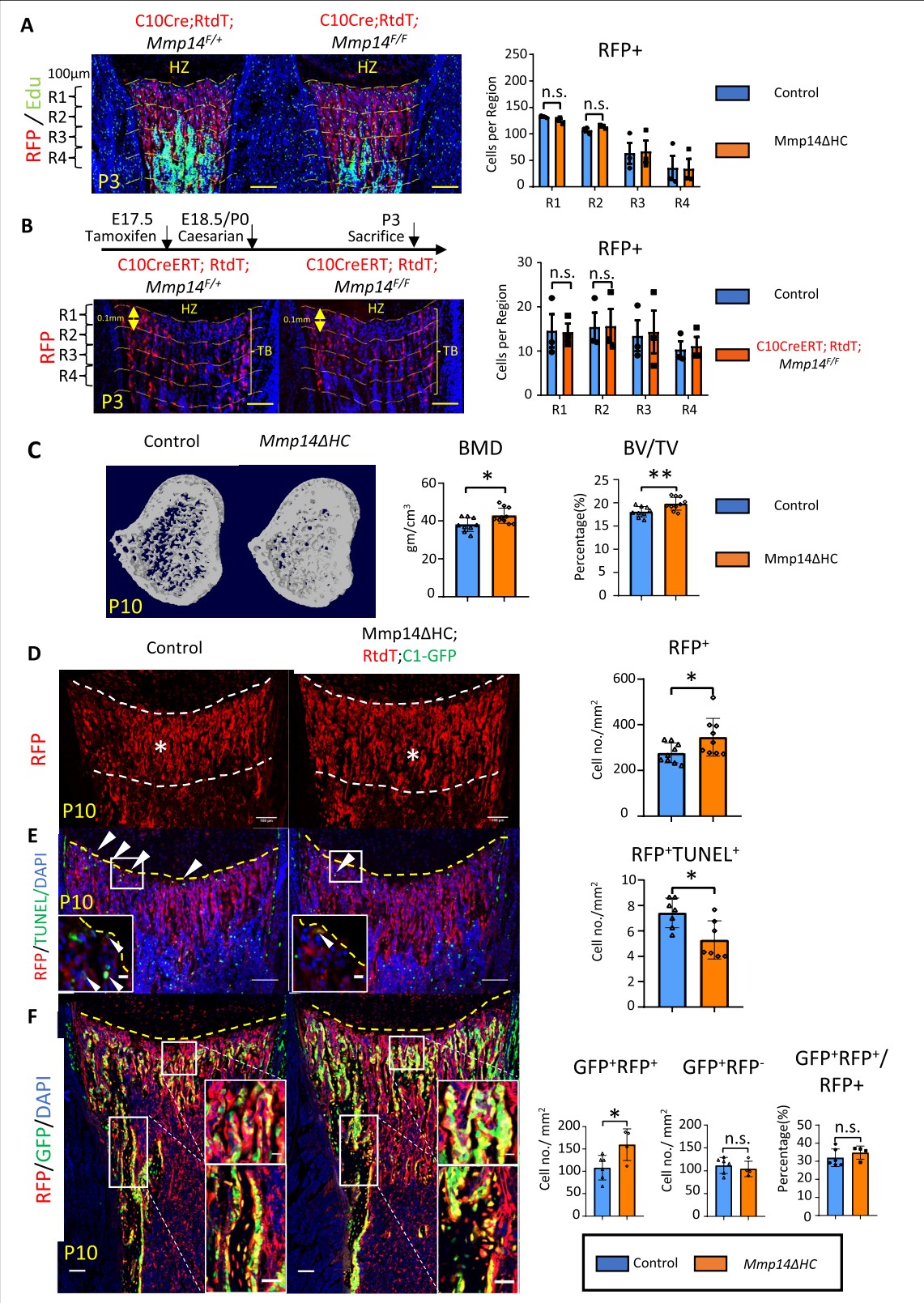

**Figure 2.** Increased osteogenesis in Mmp14ΔHC is not because of abnormal transition of hypertrophic chondrocytes (HCs). (**A**) Immunofluorescence staining of RFP and Edu-labeled cells counterstained with DAPI in Mmp14ΔHC (orange bar) and control mice tibia at P3. For quantification, the trabecular bone is divided into four zones, each 0.1 mm in thickness. The number of RFP+(red) cells in each region were counted. (**B**) Schematic of experimental design. Tamoxifen was administered to C10CreERT;Rtdt;*Mmp14^{F/+}* and C10CreERT;Rtdt;*Mmp14^{F/F}* mice at E17.5 and harvested at P3.

*Figure 2 continued*

Staining and quantification as (**A**). (**C**) Transverse image of Micro-CT analysis in Mmp14ΔHC mutants and their littermate controls at P10. Statistical analysis of bone mineral density (BMD) and bone volume over tissue volume ratio (BV/TV) using Micro-CT in Mmp14ΔHC and their littermate control (blue bar) (n = 9). The data points were pooled across multiple bone samples. (**D**) Representative immunofluorescence staining and quantification of tdTomato-labeled HC descendants in trabecular bone (asterisk) of Mmp14ΔHC and littermate controls at P10. White dotted line highlights the region below the chondro-osseous junction. Scale bars, 100 μm. The data represent means ± SEM. The data points were pooled across multiple bone samples (n = 7). (**E**), In situ terminal deoxynucleotidytransferase deoxyuridine triphosphate nick end labeling (TUNEL) assay labeling apoptotic cells and quantification in Mmp14ΔHC and control mice at chondro-osseous junction at P10 (n = 7). (**F**), Representative immunofluorescence staining and quantification of RFP (red), GFP (green) counterstained with DAPI (blue) marking tdTomato labeled HC descendants, GFP-labeled osteoblasts and nucleus in Mmp14ΔHC;Rtdt;C1-GFP and control littermates at P10. RFP+ GFP+ , RFP-GFP+ and ratio of GFP+ RFP+/RFP+ cells in the trabecular bone comparing Mmp14ΔHC and littermates. *Mmp14^F/-^*;C10Cre and *Mmp14 ^F/F^*;C10Cre were pooled and abbreviated as Mmp14ΔHC for analysis. The data points were pooled measurements across multiple bone sections per animal. n = 4 for Mmp14ΔHC, n = 6 for control. Statistics: the data represent means ± SEM. (**b, e, g, h**): p values were calculated using two-tailed unpaired *t*-tests, *p<0.05, **p<0.01, ***p<0.001.

The online version of this article includes the following figure supplement(s) for figure 2:

**Figure supplement 1.** Abnormal localization of hypertrophic chondrocytes (HC) descendants in *Mmp14^-/-^* mice.

**Figure supplement 2.** Molecular characterization of skeletal phenotype in Mmp14ΔHC mice.

(*Figure 2—figure supplement 2*). In contrast to reduced apoptotic cells, the number of Edu+-labeled proliferating cells was comparable between Mmp14ΔHC mutants and control mice (*Figure 2—figure supplement 2*).

To test if the number of osteoclasts was affected in Mmp14ΔHC mutants, we performed tartrate-resistant acid phosphatase (TRAP) staining and quantitated the number of osteoclasts in trabecular bone (*Figure 2—figure supplement 2*). In line with reports showing MMP14 deficiency does not affect osteoclastogenesis (*Tang et al., 2013*; *Zhu et al., 2020*), the number of osteoclasts was comparable between Mmp14ΔHC mutants and controls at P10, suggesting the increased number of HC-derived osteogenic cells, not reduced resorption, was a major contributor for increased bone mass at this stage. Overall, these results are in line with previous reports showing a combined inactivation of MMPs is required for osteoclast dysfunction (*Zhu et al., 2020*).

A small proportion (less than 2%) of WT HC descendants have been shown to become bone marrow adipocytes (*Yang et al., 2014b*; *Tan et al., 2020*). It was possible that changes in the collagenous microenvironment that coordinates adipogenesis could have an impact (*Chun et al., 2006*; *Sato-Kusubata et al., 2011*). We found that the total frequency of HC-derived bone marrow adipocytes in Mmp14ΔHC mice was unchanged compared with control, suggesting that MMP14 does not influence the adipogenic fate choice of HC-derived cells.

## PTH1R is a substrate of MMP14

Next, we sought to determine the underlying reason for the increased bone in Mmp14ΔHC mice. Reduced calvarial osteogenesis and osteoclast overactivity in *Mmp14* mutant mice were found attributable to cleavage of ADAM9, RANKL, and extracellular matrix proteins by MMP14, regulating FGF, RANK, and YAP/TAZ signaling (*Hikita et al., 2006*; *Chan et al., 2012*; *Tang et al., 2013*). However, these findings are insufficient to account for the increased trabecular bone observed in Mmp14ΔHC mice, suggesting a yet undiscovered mechanism could be the underlying cause of the phenotype. Interestingly, MMP14 was shown to be a downstream target of PTH signaling in osteocytes (*Delgado-Calle et al., 2018*). It has been proposed that, because cleavage of PTH1R can be inhibited by TIMP2, but not TIMP1, MMP-dependent cleavage of PTH1R causes reduced stability and degradation of PTH1R, raising the possibility that cleavage by an MMP might inhibit PTH signaling (*Klenk et al., 2010*). However, the identity of the responsible MMP was unknown. This collective evidence, including the co-expression of *Mmp14* and *Pth1r* from scRNAseq data (*Figures 1M and 3A*), strongly suggested a direct molecular link between MMP14 and PTH/PTH1R signaling pathway could be possible. Therefore, we hypothesized that MMP14 can proteolytically process and perhaps inhibit PTH/PTH1R signaling. Knocking out MMP14 in HC-derived osteogenic progeny could remove its inhibitory effect on PTH pathway.

To test the hypothesis, we assayed the impact of co-expressing (human influenza hemagglutinin) HA-tagged PTH1R with full-length MMP14 (MT1-FL) or catalytic inactive MMP14 (MT1-EA) in human 293T human embryonic kidney (HEK) cells (*Figure 3B*). After deglycosylation with

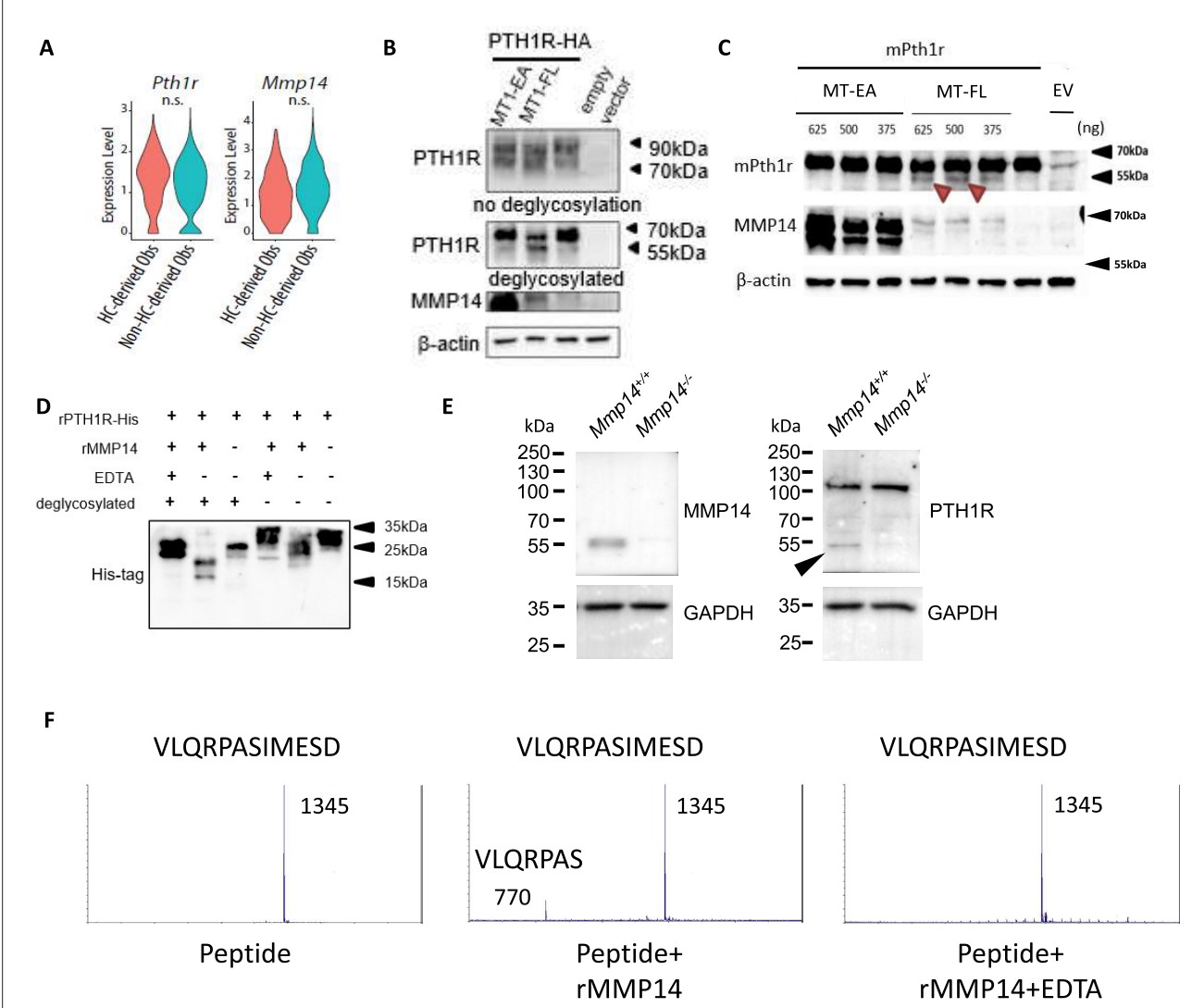

**Figure 3.** PTH1R is a substrate of MMP14. (**A**) Violin plots showing the expression levels of *Pth1r* and *Mmp14* in the hypertrophic chondrocytes (HC)-derived and non-HC-derived osteoblasts. n.s., not significant. (**B**) HEK293T cells expressing HA-tagged human PTH1R were transfected with either WT (MT1- FL) or E/A catalytic inactive mutant MMP14 (MT1 EA). β-Actin is used as a loading control. Image representative of three independent experiments. (**C**) HEK293T cells expressing mouse Pth1r (without HA-tag) were transfected with either WT(MT1-FL) or E/A catalytic inactive mutant MMP14 (MT1-EA). Mouse Pth1r was deglycosylated, detected with anti-PTH1R antibody and β-actin is used as a internal loading control. Images are representative of three independent experiments. (**D**) His-tagged rPTH1R-ECD was incubated with recombinant catalytic domain of MMP14 (lanes 1, 2, 4, 5) with EDTA (lanes 1, 4), and with (lanes 1–3) and without (lanes 4–6) deglycosylation. Western blotting analyses used specific antibodies as indicated. Data are representative of three independent experiments. His-tagged rPTH1R-ECD was incubated with recombinant catalytic domain of MMP14 at two enzyme/substrate ratios (rPTH1R-ECD only, 1:50, 1:10, and 1:10 with EDTA). The protein mixture were deglycosylated (left panel). (**E**) Western blots showing cleaved fragment of PTH1R in trabecular bone extracts from *Mmp14*[+/+](left lane) and *Mmp14*[-/-](right lane) mice at P14. (**F**) Synthetic peptide from extracellular domain of PTH1R 55–67 was incubated with recombinant MMP14. The presence of 770 Da suggests a fragment of VLQRPAS.

The online version of this article includes the following source data and figure supplement(s) for figure 3:

**Source data 1.** Contains raw uncropped western blot gel photos of *Figure 3* and *Figure 3—figure supplement 1*.

**Figure supplement 1.** MMP14 is a major protease for PTH1R.

PNGaseF, western blotting showed, in the presence of MT1-FL, a truncated form of PTH1R-HA with size close to 55 kDa is increased compared to empty vector and MT1-EA (*Figure 3B*). To exclude that the cleavage is an artifact due to HA-tag insertion in the extracellular domain, a mouse *Pth1r* cDNA without HA-tag was reverse transcribed from extracted RNA and cloned into pcDNA3.1 expression vector (*Figure 3C*). Like PTH1R-HA, a truncated protein fragment of PTH1R

was detected when MT1-FL was expressed, confirming proteolytic processing of PTH1R exists both in mice and human (*Figure 3C*). To exclude the possibility of indirect cleavage of PTH1R by MMP14 in 293T cells, a recombinant extracellular domain of human PTH1R with polyhistidine tag at C-terminus (PTH1R-ECD, amino acid 1–181) was incubated with and without recombinant human MMP14 (rhMMP14) overnight (*Figure 3D*). Western blots showed, compared to full-length deglycosylated PTH1R-ECD, after incubation with rhMMP14 (lane 2), there appeared at 25 kDa (lane 3), two truncated fragments with size around 20 kDa and 15 kDa (*Figure 3D*). These results suggest that rhMMP14 directly cleaves PTH1R-ECD at multiple sites, with one cleavage site at around amino acid 60 and another at around 90–100, accounting for the two histidine-tagged fragments detected by SDS-PAGE observed (*Figure 3D*). Blocking MMP14 activity by EDTA inhibited PTH1R cleavage (*Figure 3D*). To test if other MMPs/ADAMs participate in the cleavage of PTH1R, we assayed ADAM10, ADAM15, and ADAM17 and found none of them cleaved (*Figure 3—figure supplement 1A*). Co-immunoprecipitation of PTH1R-HA and MT1-FL suggested that MMP14 can interact with PTH1R (*Figure 3—figure supplement 1B and C*).

To test if MMP14 cleavage of PTH1R occurred in vivo, trabecular bone protein lysate was prepared from wild-type and *Mmp14*-/- mice at P14. A truncated 55 kDa form of PTH1R can be detected in the *Mmp14*+/+ trabecular bone lysates but not in those from *Mmp14*-/- mutants suggesting MMP14 may be the sole protease/MMP that cleaves PTH1R in osteogenic cells (*Figure 3E*). However, an additional protein band at 110 kDa was observed in in vivo-derived samples compared to secondary cell lines, possibly reflecting differences in post-translational modification (*Lackman et al., 2018*). The scRNAseq profiling at P56 suggests a portion of cells co-express *Pth1r*, *Mmp14*, GFP, and WPRE transcript, consistent with a role of MMP14 as a protease for PTH1R (*Figure 1M*, *Figure 1—figure supplement 1D*, *Figure 3—figure supplement 1D*).

Given that rhMMP14 directly cleaves PTH1R-ECD into fragments with size around 20 kDa and 15 kDa, we propose at least one putative cleavage site exist around amino acid 55–65 in PTH1R. Computational prediction also suggest a possible cleavage site exist at amino acid around 61 (*Kumar et al., 2015*). To test this, we incubated rhMMP14 with a synthetic peptide with amino acid sequence 55–67 from PTH1R. Using mass spectrometry, we found a peak at 770 Da consistent with molecular mass of a VLQRPAS fragment, suggesting that amino acid 61 is one of the cleavage sites of MMP14 (*Figure 3F*). Taken together, rhMMP14 cleaves PTH1R-ECD at multiple sites in vitro, suggesting two possible fragments, PTH1R (61–593, predicted molecular size 59 kDa) or PTH1R (100~593, predicted molecular mass 55 kDa) are possible, and therefore, is consistent with the truncated PTH1R peptides observed in vivo (~55 kDa).

## MMP14 inhibits PTH/PTH1R signaling

Prior molecular and structural studies have demonstrated that the exon 2 encoding region of PTH1R, which harbors the cleavage site by MMP14, is dispensable for both binding to PTH/Pthrp and receptor function (*Lee et al., 1994*; *Lee et al., 1995*; *Zhao et al., 2019*; *Ehrenmann et al., 2018*). Since MMP14 has a wide range of substrates ranging from collagens to transmembrane ligands, we hypothesized that cleavage of MMP14 promoted the degradation of PTH1R. To explore the molecular consequences of MMP14 cleavage on PTH1R, 293T cells stably expressing PTH1R transfected with MT1-FL or MT1-EA were challenged with PTH and their responses examined by calculating the relative amounts of phospho-CREB (p-CREB) (*Figure 4A and B*). As expected, lower relative amounts of p-CREB, p-ERK, and cyclic-AMP (cAMP) were observed after PTH challenge in 293T cells expressing MT1-FL compared to control, suggesting that MMP14 inhibits PTH signaling by facilitating receptor degradation (*Figure 4B and C*). We isolated trabecular osteoblasts from *Mmp14*-/- mice in order to ensure only MMP14-deficient cells were analyzed. These MMP14 deficient primary trabecular osteoblasts showed increased p-CREB activity in response to PTH (*Figure 4D*). We titrated the level of MT1-FL against the same amount of PTH1R and found increasing MT1-FL further promoted PTH1R degradation (*Figure 4—figure supplement 1A*). Consistent with previous reports, MMP14 is upregulated in response to PTH treatment (*Figure 4—figure supplement 1B*). Overall, these results identify cleavage of PTH1R by MMP14 at amino acid 61 as a novel mechanism for modulating GPCR stability and subsequently attenuated PTH signaling (*Figure 4E*, *Figure 4—figure supplement 1*).

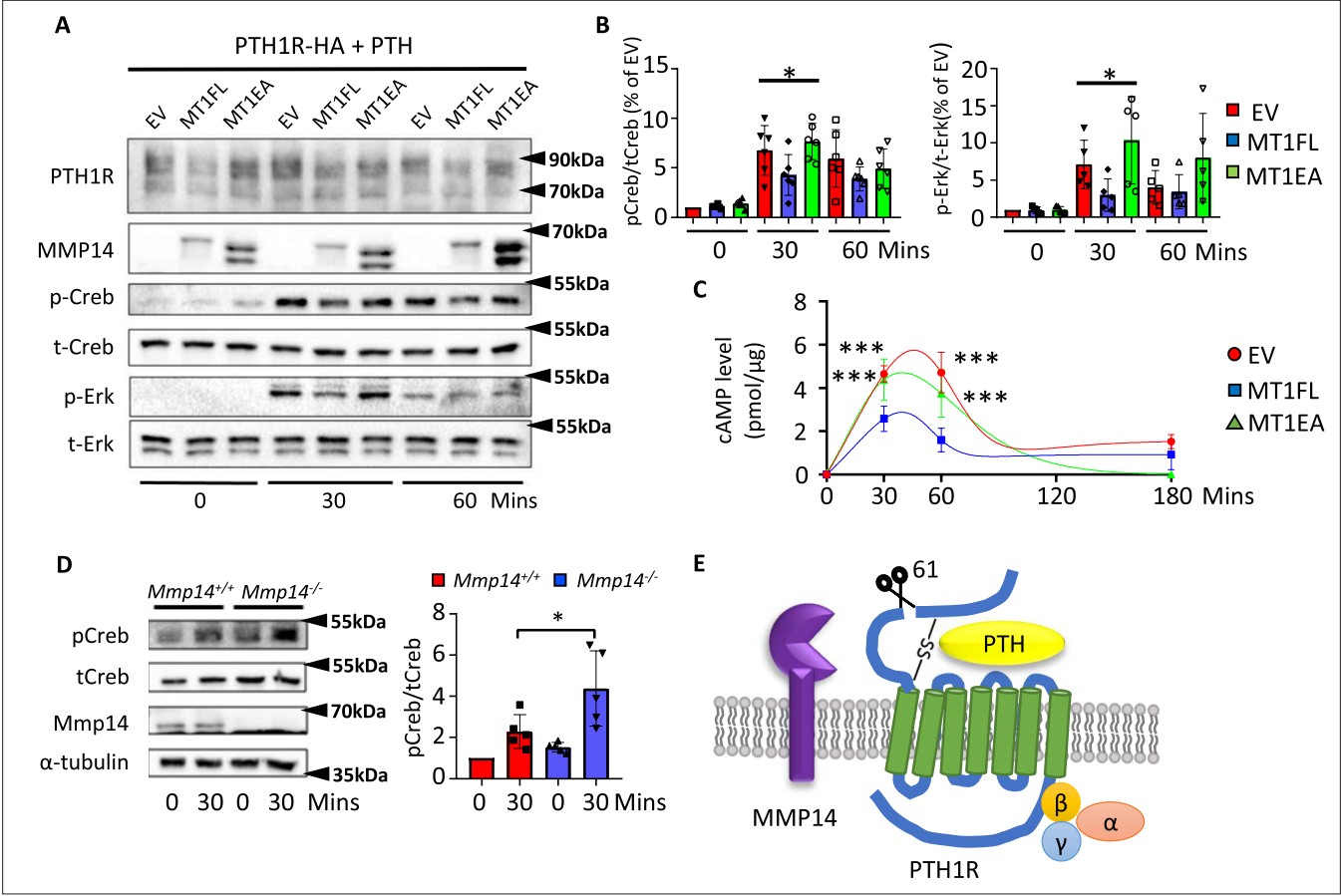

**Figure 4.** MMP14 inhibits PTH signaling. (**A**), MMP14 can inhibit PTH signaling in HEK293 cells. HEK293 cells stably expressing PTH1R-HA were transfected with empty vector (EV), MMP14 (MT1-FL), and catalytic inactive MMP14 (MT1-EA). Cells were challenged with $1 \times 10^{-7}$M PTH in vitro for 0, 30, and 60 min. PTH1R, MMP14, phospho-CREB (p-CREB), total-CREB (t-CREB), phosphor-Erk, total-Erk, and loading control β-actin were analyzed by western blotting. (**B**) Relative level of p-CREB/t-CREB and p-Erk/t-Erk was quantitated and measured using ImageJ (n = 6 for p-CREB/t-CREB and n = 5 for p-Erk/t-Erk). One-way ANOVA followed by unpaired *t* Welch's correction was used to determine statistical significance. Data are presented as means ± SEM. **p<0.01, *p<0.05. (**C**) HEK 293 cells expressing PTH1R were challenged with PTH for 30, 60, and 180 min and the level of cAMP with MT1-FL, MT1-EA, or empty vector were measured (n=5). One-way ANOVA followed by unpaired *t* Welch's correction was used to determine statistical significance. (**D**) Trabecular osteoblasts extracted from *Mmp14+/+* (red bar) and *Mmp14-/-* (blue bar) mice were challenged with $1 \times 10^{-7}$ M PTH(1–34). Cell lysate were analyzed by western blotting for p-CREB and t-CREB level (left). Right: quantification of relative level of p-CREB/t-CREB in trabecular osteoblasts extracted from *Mmp14+/+* and *Mmp14-/-* mice (n = 5). Data are presented as means ± SEM. **p<0.01, * p<0.05, unpaired *t*-test. (**E**) Schematic diagram showing MMP14 cleaves PTH1R at amino acid 61. Seven-pass transmembrane domains are in green. PTH1R transduces signals via G-protein complex α, β, γ.

The online version of this article includes the following source data and figure supplement(s) for figure 4:

**Source data 1.** Contains raw uncropped western gel photos for *Figure 4* and *Figure 4—figure supplement 1*.

**Figure supplement 1.** MMP14 destabilize PTH1R and can be upregulated by PTH.

## HC-derived osteogenic cells respond to PTH, which is enhanced in Mmp14ΔHC

Having demonstrated MMP14 can directly cleave PTH1R to negatively regulate its function, we next asked if PTH stimulates the osteogenesis of HC-derived cells and if MMP14 moderates PTH signaling in HC-derived cells in vivo. While treatment of mice with PTH inhibits apoptosis of osteoblast precursors and promotes their differentiation into mature osteoblasts (*Balani et al., 2017*), whether HC-derived osteoblasts respond and contribute to the anabolic response to PTH reported for osteoblasts is not known. To assess the contribution of HC-descendants to C1-GFP-expressing mature osteoblasts induced by PTH treatment, the C1-GFP transgene was introduced into C10Cre;Rtdt and Mmp14ΔHC;Rtdt mice to label osteoblasts in bone (*Figure 5A*). We examined the skeletal phenotype

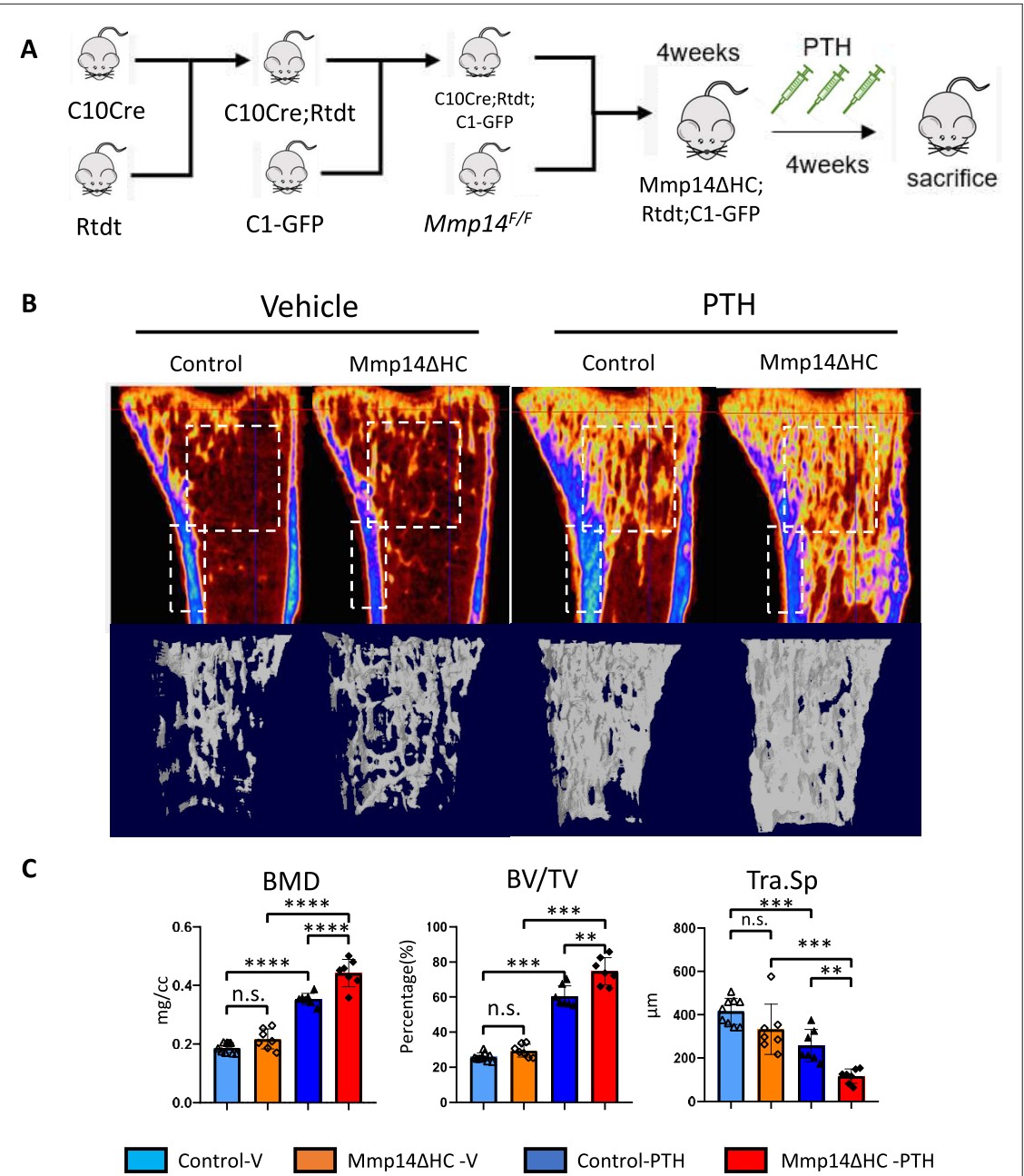

**Figure 5.** Increased anabolic response of hypertrophic chondrocyte (HC)-derived osteoblasts to PTH treatment in Mmp14ΔHC mice. (**A**) Schematic diagram showing experimental design for generation of Mmp14ΔHC;Rtdt;C1-GFP mice for teriparatide treatment. (**B**) Sagittal and reconstructed 3D image of micro-CT of control and Mmp14ΔHC mice treated with vehicle and PTH(1–34) for 1 month starting from P28 for five injections per week. (**C**) Statistical analysis of bone mineral density (BMD), bone volume over tissue volume ratio (BV/TV), and trabecular separation (Tra.Sp.) using Micro-CT in control (blue [n = 9] and deep blue [n = 7] bars) and Mmp14ΔHC (orange [n = 7] and red [n = 7] bars), mice treated with vehicle (blue and orange) and PTH(1–34) (deep blue and red), for 1 month. Data are presented as means ± SEM. **p<0.01, *p<0.05, One-way ANOVA followed by unpaired *t* Welch's correction was used to determine statistical significance. Control samples were pooled from mice of following genotypes: *Col10a1^{Cre/+};Mmp14^{F/+};Rosa26^{LSL-tdTomato/LSL-tdTomato}*;C1-GFP and *Col10a1^{Cre/+};Mmp14^{F/+};Rosa26^{LSL-tdTomato/LSL-tdTomato}* and *Col10a1^{+/+};Mmp14^{F/+};Rosa26^{LSL-tdTomato/LSL-tdTomato}*;C1-GFP. Mmp14ΔHC samples were from *Col10a1^{Cre/+};Mmp14^{F/-};Rosa26^{LSL-tdTomato/LSL-tdTomato}*;C1-GFP mice.

in Mmp14ΔHC mutants treated with PTH for 4 weeks (***Figure 5B***). Compared to PTH stimulation in WT controls, PTH treatment in Mmp14ΔHC mutants caused a 15 and 21% further increase in BMD and BV/TV, respectively (***Figure 5B and C***). PTH treatment dramatically increased the amount of HC progeny (tdT⁺, detected by RFP immunostaining) in Mmp14ΔHC trabecular bone (***Figure 5B and C***).

In compound C10Cre;Rtdt;C1-GFP mice, mature osteoblasts, pre-osteoblasts, and HC descendants were labeled by GFP, OSTERIX, and RFP, respectively. We found many more HC descendants in the trabecular region that co-expressed OSTERIX, a marker of osteoprogenitors, and C1-GFP in response to PTH treatment than in controls (*Figure 6A and B*). These HC descendants showed positive pCREB⁺RFP⁺ staining confirming PTH signaling activity in HC derivatives (*Figure 6C*). HC-derived cells constituted approximately 50% of all trabecular osteoblasts with and without PTH treatment (*Figure 6D*). HC-derived osteoblasts therefore represent a significant osteogenic population directly contributing to half of the total osteoblast population and capable of responding to PTH treatment in the same way as non-HC-derived osteoblasts (*Figure 6D*). In Mmp14ΔHC mice, the number of RFP+ and RFP+ GFP+ and RFP+ OSX+ cells further increased compared to control mice treated with PTH, consistent with the heightened response to PTH treatment in *Mmp14*-deficient HC-derived cells (*Figure 6D*).

By contrast with enhanced osteogenesis in trabecular bone, BV/TV and cortical BMD were comparable between PTH-treated Mmp14ΔHC mutants compared to PTH-treated controls (*Figure 6—figure supplement 1*). There was no significant increase in RFP+ GFP+ HC-derived endosteal cells in Mmp14ΔHC mutants. In the HC-derived (RFP+ GFP+ ) endosteal cell fraction, the response to PTH was enhanced in mutants, suggesting differences in the effect of MMP14 deficiency and PTH sensitivity of HC-derived endosteal cells compared to the non-HC-derived counterparts. However, CyclinD and Runx2 showed increased expression in Mmp14ΔHC mutants, suggesting a relationship between PTH signaling to osteogenesis and cell cycle regulation in HC-derived cells (*Figure 6—figure supplement 1C*).

Although the number of HC-derived RFP+ osteocytes were also not different between controls and mutants, there was a significantly enhanced response of HC-derived osteocytes to PTH in the cortical bone of Mmp14ΔHC mutants (*Figure 6—figure supplement 1A and B*). These results are consistent with reports showing increased cortical bone in human osteoporotic patients with PTH treatment (*Hansen et al., 2013*; *Tsai et al., 2016*). However, this is in contrast to the observation that PTH treatment results in increased trabecular bone in patients but had no effect on endo-cortical, cortical, or periosteal bone, or that intermittent treatment of PTH in mouse starting from P28 does not fully replicate PTH treatment in human adults (*Recker et al., 2009*). Whether the discrepancy in outcome of PTH treatment or MMP14 deficiency on cortical bone and trabecular bone is related to a more transient effect on PTH signaling on bone synthesis in the former than in the latter and/or remodeling differences are questions for future study.

## HC descendants persist and contribute to the anabolic response to PTH in aged mice

Although, unlike in humans, the growth plate of mice does not close, in adult mice by 1 year of age, the hypertrophic cartilage is vestigial and endochondral ossification diminishes substantially (*Haseeb et al., 2021*). To test whether the HC-derived osteogenic cells could also contribute to the response to PTH treatment in older mice, we treated 1-year-old C10Cre;Rtdt mice with PTH for 4 weeks (*Figure 7A–Ci–iii*). Like in young mice, HC descendants at 1 year old responded to PTH(1–34) treatment by producing more osteogenic progeny (*Figure 7Ci–iii*), suggesting sustained ability of HC-derived osteoblasts to contribute to PTH-stimulated osteogenesis with aging. These tdTomato-positive progenies also express MMP14 (*Figure 7—figure supplement 1A*). Taken together, our findings implicate a molecular link between MMPs and PTH signaling in regulating the activity of HC-derived osteoblasts in postnatal and adult mice (*Figure 8*).

## Discussion

Recent studies have established the concept that in the growth plate, resting chondrocytes function as skeletal stem cells for continuous supply of proliferating and pre-hypertrophic chondrocytes and HCs (*Mizuhashi et al., 2018*; *Newton et al., 2019*). These cells form part of a continuum in which the HCs transition into the full osteogenic lineage, contributing to the formation of primary spongiosa and trabecular bone (*Tsang and Cheah, 2019*; *Yang et al., 2014a*; *Yang et al., 2014b*; *Park et al., 2015*; *Tan et al., 2020*) and participating in bone regeneration and healing (*Hu et al., 2017*; *Yang et al., 2014a*; *Yang et al., 2014b*; *Matsushita et al., 2021*). Here we show that chondrocyte-derived

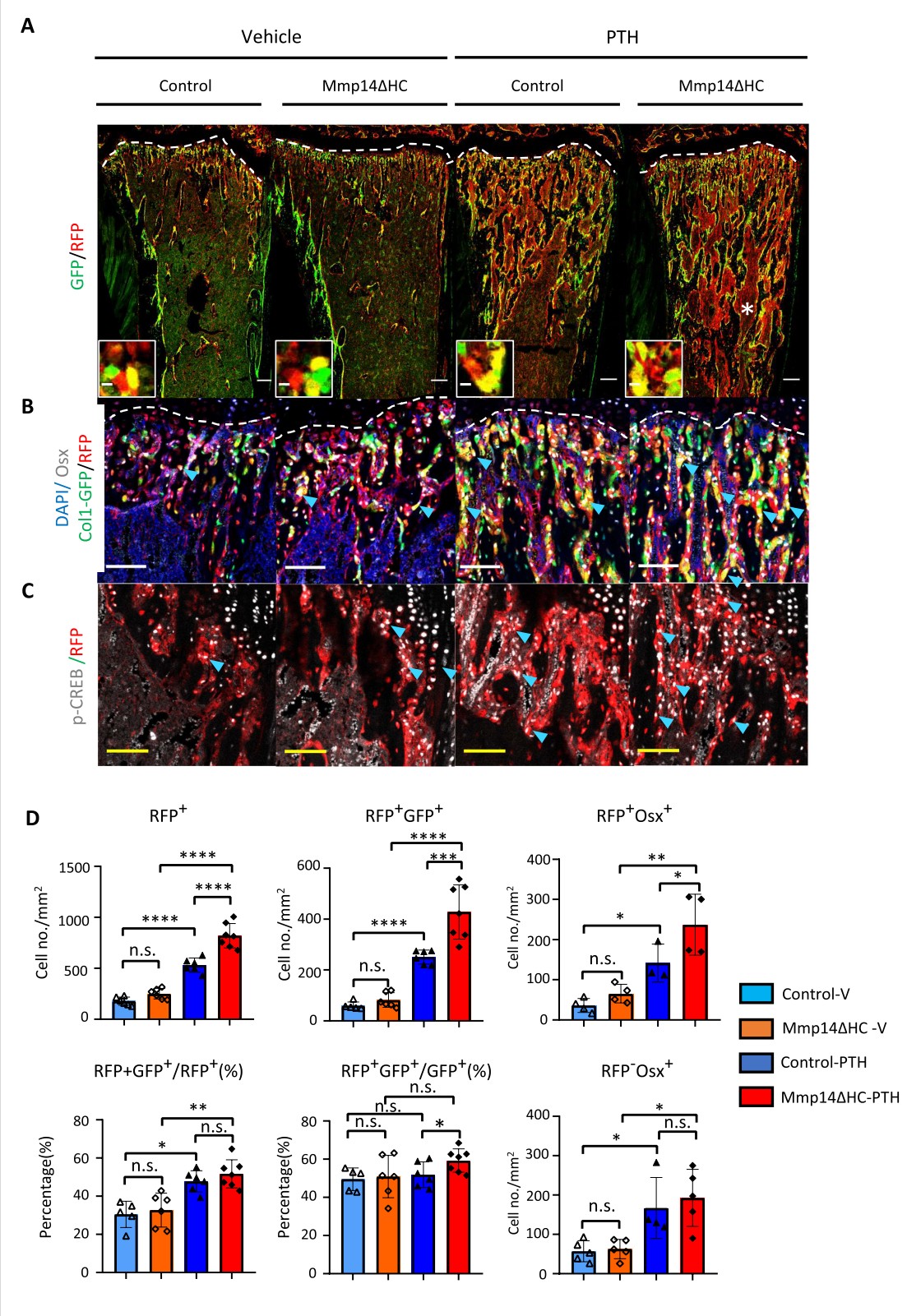

**Figure 6.** MMP14 controls expansion of chondrocyte descendants after PTH treatment. (**A**) Representative immunofluorescence staining of GFP (green) and RFP (red), marking hypertrophic chondrocytes (HC)-descendants and osteoblasts, respectively. Mmp14ΔHC mutant mice (8 weeks) and their littermate controls were treated with vehicle or PTH(1–34) for 4 weeks. Yellow dotted line demarcates chondro-osseous junction. Scale bar 50 μm. (**B**) Fluorescence colours mark nuclei, pre-osteoblasts, and HC-descendants as for *Figure 2*. Scale bar 100 μm. (**C**) Representative immunofluorescence

*Figure 6 continued on next page*

*Figure 6 continued*
staining of phospho-CREB (pCREB [gray] and RFP [red]). Scale bar 100 μm. (**D**) Quantification of cell number/mm²: RFP+ (HC descendants); RFP+ GFP+ (mature HC-derived Osteoblasts); RFP+ Osx+ (HC-derived pre-osteoblasts), ratio of RFP+ GFP+ /RFP+ cells, ratio of RFP+ GFP+/GFP+ cells, number of RFP+ Osx+ and number of RFP-Osx+ cells in control (blue and deep blue) and Mmp14ΔHC (orange and red) mice (n = 5). The data represent means ± SEM. p-Values were calculated using one-way ANOVA followed by unpaired *t* Welch's correction. Statistical significance of RFP+GFP+/GFP+ ratio is analyzed by Student's *t*-test (*p<0.05). The data points were pooled across multiple bone samples.

The online version of this article includes the following figure supplement(s) for figure 6:

**Figure supplement 1.** Differential response to PTH in cortical osteocytes compared to trabecular osteoblast.

osteoblasts contribute a significant proportion to osteogenesis arising from PTH-mediated stimulation of bone formation. The data implicate hypertrophic chondrocytes as a reservoir of progenitors supplying osteoblast precursors and this process is controlled by MMP14 modulation of PTH signaling by direct cleavage of PTH1R. MMP14 inhibits PTH1R signaling and genetic targeting of MMP14 boosts PTH-mediated bone synthesis.

Our study has provided new in vivo insights into the properties and functional contribution of chondrocyte-derived osteoblasts to maintaining normal bone mass in response to PTH. The single-cell transcriptomic analyses and lineage tracing in mouse models show that HC-derived cells express

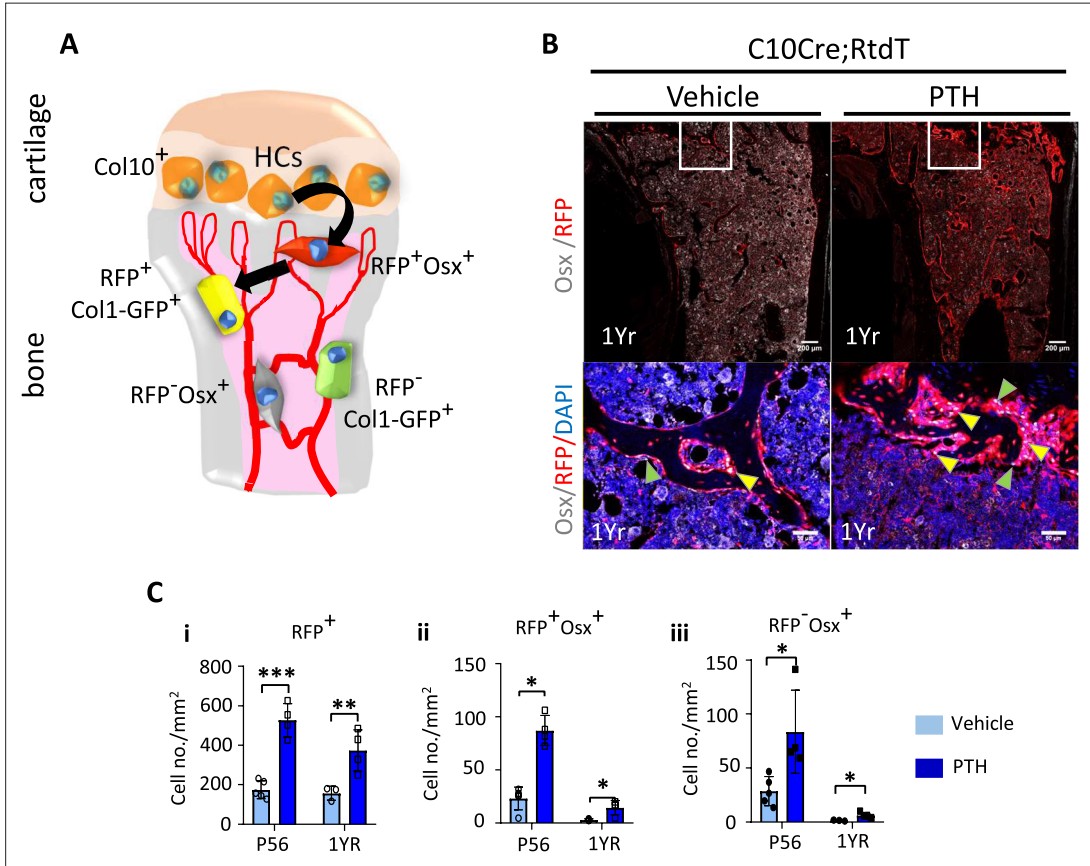

**Figure 7.** Hypertrophic chondrocyte (HC) descendants persists in trabecular bone and contribute to PTH response in ages adult mice. (**A**) Schematic diagram showing ontogeny of HC-derived and non-HC-derived osteogenic cells and their markers. (**B**) Immunofluorescence staining of DAPI (blue), Osterix (gray), and RFP (red) for marking cell nuclei, pre-osteoblasts, and HC-descendants, respectively. Mice were treated with vehicle or PTH(1–34) for 4 weeks at five injections per week starting from 1 year old. Magnified picture of at the trabecular bone (white box) is also presented. Yellow arrows mark HC-derived pre-osteoblasts. (**C**) P56 and 1-year mice treated with PTH: (i–iii) quantification of number/mm² of RFP+ cells, RFP+ Osx+ cells, RFP-Osx+ cells (n = 3). Data are represented as mean ± SEM. p-Values were calculated using two-tailed unpaired *t*-test, *p<0.05, **p<0.01, ***p<0.001. Data points were pooled measurements across multiple bone sections per animal.

The online version of this article includes the following figure supplement(s) for figure 7:

**Figure supplement 1.** Expression of *Mmp14* in hypertrophic chondrocyte (HC)-derived cells in 1-year-old mice.

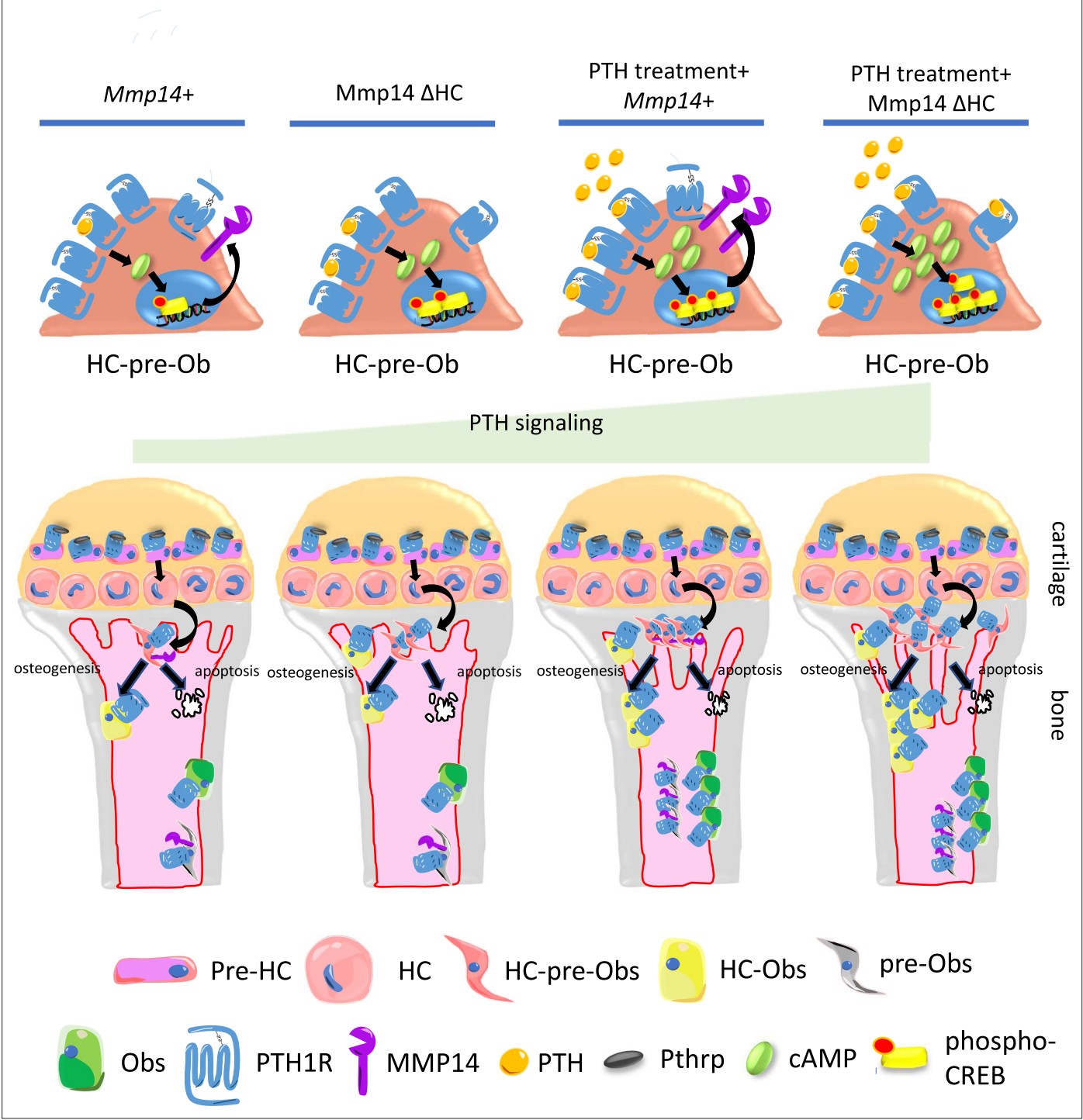

**Figure 8.** Model of MMP14 moderation of PTH signaling intensity and osteogenesis. Upon PTH binding, PTH1R generates cAMP signals to activate cAMP response element binding protein (CREB). Activation of PTH1R causes upregulation of MMP14. MMP14 in turn cleaves PTH1R to inhibit its signaling activity. In the chondrocyte lineage, pre-hypertrophic chondrocytes (pre-HC) differentiate to hypertrophic chondrocytes (HCs), translocate to the subchondral space, and subsequently to HC-pre-osteoblasts contributing to trabecular, endosteal and endocortical bone and can respond to PTH treatment. Together, MMP14 moderates the HC derivatives' response to PTH ensuring a controlled supply of osteoblast precursors and thereby bone anabolism. Obs, osteoblasts; PHZ, hypertrophic zone; HZ, hypertrophic zone; TB, trabecular bone.

genes characteristic of skeletal progenitors, mesenchymal cells, immature and mature osteoblasts, and terminally differentiated osteocytes even at maturity (P56) in trabecular and endosteal/endocortical bone. Despite this gross similarity, there were a few genes that were differentially expressed between C10Cre-labeled HC-derived and non-HC-derived osteoblasts, within the primary spongiosa, the functional significance of which is unknown and worthy of future study.

In our study, we have found, by contrast to the decreased trabecular bone caused by global knockout of *Mmp14*, that conditionally inactivating MMP14 in the HC-derived lineage results in a modest increase in bone at P10 by promoting their survival. Our findings identify MMP14 as a key protease that processes PTH1R, regulating the response of HC derivatives to PTH and thereby their contribution to trabecular bone. MMP14 inhibits osteoclastogenesis via controlled release of soluble RANKL (*Delgado-Calle et al., 2018*). RANKL has been reported to be a downstream target of PTH and a substrate of MMP14, which implies that MMP14 is downstream of PTH signaling, but the exact molecular relationship is unclear. Notably osteoclastogenesis was normal in the P10 Mmp14ΔHC mutants, suggesting that the increased bone mass is intrinsic to HC-derived osteoblasts and unrelated to an impact on the production of soluble RANKL in the absence of MMP14. Because inactivating MMP14-mediated intracellular signaling constitutively by mutating the cytoplasmic tail of MMP14 leads to increased trabecular bone and decreased adipocytes in mice, it has been suggested that MMP14 regulates the osteogenic versus adipogenic cell fate decision of differentiating bone marrow skeletal stem cells (*Attur et al., 2020*). However, there was no effect on the frequency of HC-derived adipocytes in Mmp14ΔHC mutants, suggesting that the increased osteogenesis is not caused by a change in HC cell fate.

GPCR is the largest family of transmembrane proteins with contributing to ~35% of encoded proteins. Around 3–40% of drugs are derived to target GPCRs. Few preliminary reports propose proteolytic modification of GPCRs by MMPs or ADAMs might modulate their signaling activity (*Lackman et al., 2018*; *Goth et al., 2017*). A recent paper proposed that cleavage of PTH1R could regulate its downstream signaling (*Klenk et al., 2022*). Our study has uncovered that PTH1R, a class-B G-protein coupled receptor, is a novel direct substrate for MMP14, which, MMP14, by cleaving PTH1R acts to inhibit PTH1R signaling. PTH1R was previously found to be regulated by different post-translational modifications, including phosphorylation and glycosylation (*Qiu et al., 2010*; *Izumida et al., 2020*). The detailed structure of PTH1R and the lack of functional impact of HA-tagging in this domain suggest the unstructured region encoded by exon 2 does not participate in receptor-ligand binding nor signal transduction (*Zhao et al., 2019*; *Ehrenmann et al., 2018*; *Couvineau et al., 2008*). Our results demonstrate that proteolytic modification of this region directly influences GPCR signaling via GPCR stability. We propose a novel paradigm of MMP14 activity in which it also mediates direct titration of PTH signaling in the HC-osteoblast lineage (*Figure 8*). In this model, the intensity of PTH stimulation of osteogenesis is curbed via MMP14 cleavage of the ectodomain of PTH1R. Hence, inactivation of MMP14 in HC descendants spares the ectodomain of PTH1R from cleavage, resulting in an increased anabolic response to PTH and therefore places MMP14 also upstream of PTH signaling. Since MMP14 is upregulated by PTH signaling in osteogenic cells in vivo (*Delgado-Calle et al., 2018*), it is possible that MMP14 is also involved in a negative feedback protection mechanism to constrain osteogenesis in response to PTH challenge. Given that other MMPs or ADAMs have been shown to participate in transmembrane proteins cleavage, whether MMPs other than MMP14 are involved in this process and whether PTH1R undergoes additional proteolytic modification remains to be solved in future. Interestingly, in humans several mutations, including a G100->D mutation in the ectodomain of PTH1R, has been associated with Olliers disease. Whether any of these mutations cause abnormal MMP14 cleavage of PTH1R should be verified in the future (*Couvineau et al., 2008*).

Maintaining appropriate amount of bone throughout life, mediated by hormones such as PTH, is critical for healthy aging and quality of life. However, there is a gap in knowledge on the specific contributions of individual cell types to maintaining homeostasis in bone because of the myriad of signaling pathways and diverse cell-type responses. The importance and need of a molecular and functional definition of the contribution of specific cell types in bone are illustrated by the long-term systemic treatment of osteoporosis by teriparatide (PTH 1–34). The treatment is complicated by side effects because of the physiological and pleiotropic effects of PTH signaling on different cells in the whole body. We have shown that HC-derived osteoblasts contribute to the anabolic activity in bone, responding to PTH in the same way as their non-HC-derived counterparts, and constitute a significant

proportion of the osteogenic precursors stimulated by teriparatide (PTH 1–34) treatment, even in adult mice. This response to PTH occurs in early postnatal (P10) and mature mice (1 year). Questions for future study are whether the response is as robust in aged mice (2 years or older) and whether, compared to other sources of osteoblasts, the history of PTHRP/PTH1R signaling in pre-hypertrophic chondrocytes (*Mizuhashi et al., 2018*) produces a molecular memory that is advantageous for the ability of HC descendants to mount a response to PTH. Intermittent treatment with PTH/teraparatide also induces bone resorption leading to an 'anabolic window' of bone mass gain that restricts long-term use of this treatment strategy (*Hattersley et al., 2016*; *Sharma et al., 2022*). It would be important in future to determine the long-term impact of increased PTH signaling on bone anabolism and physiology in Mmp14ΔHC mice as they age. Other questions that should be followed up are whether a defect in the HC-lineage progression underlies the increase in trabecular bone in *Mmp13* global knockouts and *Col1a1*-cre mediated conditional *Mmp13* mutants (*Stickens et al., 2004*), and the exact underlying molecular mechanism(s). MMP14 controls other signaling pathways such as FGF signaling and also YAP/TAZ signaling (*Hikita et al., 2006*; *Chan et al., 2012*; *Tang et al., 2013*). It would also be important to identify additional pathways and substrates affected by MMP14 deficiency in the osteogenic lineage, and to what degree such impact contributes to the overall changes in bone anabolism. Further, it would be important to determine whether there is a causal link between the MMP14 deficiency in the chondrocyte-osteoblast lineage and human bone-wasting disorders.

Our study illuminates understanding of the relative contribution of different cell types to maintaining bone homeostasis and highlights the importance of HCs as a source of pre-osteoblasts capable of rapid response to PTH treatment. The minor differences in transcriptomic signature between HC-derived and non-HC-derived osteoblasts and their co-expression of *Mmp14* and *Pth1r* suggest that the same paradigm of MMP14-mediated titration of PTH signaling applies in all osteoblasts. Addressing this possibility purely in non-HC-derived osteoblasts is hampered by the lack of appropriate cre reagents that can target osteoblasts in the perichondrium specifically and the intrinsic skeletal defects in Osterix–cre mice (*Huang and Olsen, 2015*). The discovery that MMP14 titrates the response to PTH and bone anabolism in vivo, contributes new insights into the mechanism of adjusting PTH control of osteogenesis and bone homeostasis, and has significance for understanding bone metabolism with implications for the development of treatments for bone-wasting diseases such as osteoporosis and in overcoming the side effects of teriparatide (PTH 1–34). Our work also contributes to understanding the scientific basis of PTH therapy.

### Ideas and speculation

Our findings provide ideas for potential therapeutic enhancement of treating skeletal diseases. It has been suggested that a therapeutic approach could be to reduce bone resorption by targeting the myeloid lineage that would require inactivating both MMP9 and MMP14 (*Zhu et al., 2020*). By contrast, a pan osteoblast-specific or HC-derived osteoblast-specific targeted manipulation approach could have a significant advantage as MMP14 activity (and therefore PTH signaling) only needs to be manipulated in a single cell type. Perhaps such an approach could avoid the need for systemic intermittent injection of PTH and thereby ameliorate side effects.

## Materials and methods
### Genetically modified mouse strains

*Mmp14*$^{-/-}$ and *Mmp14*$^{flox/flox}$ mice were generated as described previously (*Zhou et al., 2000*; *Jin et al., 2011*). *Col10a1*-Cre and *Col10a1*-CreERT mouse lines are described in *Yang et al., 2014a*. *Col1a1*--GFP and *Rosa26* reporter transgenic mice were obtained as previously described (*Kalajzic et al., 2002*; *Srinivas et al., 2001*). *Col1a1*- GFP (abbreviated as C1-GFP) transgenic mice were made by directed expression of GFP by a 2.3 kb rat type I collagen fragment (*Kalajzic et al., 2002*). PTH delivery to mice was performed as described in *Balani et al., 2017*. In brief, PTH(1–34) (acetate, Bachem) was dissolved in 0.01 M acetic acid, 150 nM NaCl, and subcutaneously injected to mice at 400 ng/g body weight for 5 days per week (*Balani et al., 2017*). *Mmp14*$^{F/-}$;C10Cre (abbreviated as *Mmp14*$^{-/ΔHC}$) and *Mmp14*$^{F/F}$;C10Cre (abbreviated as *Mmp14*$^{ΔHC/ΔHC}$) were pooled and collectively referred to as Mmp14ΔHC for analysis. All animal experiments were approved by the Institutional Animal Care and Use Committee of the University of Hong Kong and performed in accordance with

guidelines of the Committee on the Use of Live Animals for Teaching and Research of the University of Hong Kong: Protocol nos: 3981-1, 5295-20, and 5527-20.

## Isolation of cells for single-cell transcriptomics

Single cells from the growth plate and trabecular bone were harvested, disassociated, digested, and sequenced as follows. For neonatal (P6) mice, cells were isolated from whole tibia (n = 2) including the growth plate (except secondary ossification center at both ends) of a C10Cre;*Irx3*$^{+/\Delta HC}$*Irx5*$^{+/-}$; *Rosa26*$^{LSL-tdTomato/+}$ heterozygote previously shown to have normal bone mass (*Tan et al., 2020*). The tibiae (n = 2) were diced into fragments and digested with digestion solution (0.25% dispase, 0.25% collagenase type II in HBSS; Sigma-Aldrich, Cat# H6648) (5 ml in total) at 37°C on a shaker for 1 hr. To prepare cells from adult P56 mice (C10Cre;*Mmp14*$^{+/F}$;*Rosa26*$^{LSL-tdTomato/LSL-tdTomato}$;C1-GFP) (n = 3), trabecular bone and part of the growth plate from tibia were harvested. Osteogenic cells were isolated by a modification of protocol described by *Chan et al., 2012*. In brief, the bone samples were diced into fragments and digested with digestion solution. The first two suspensions obtained after 20 min sequential digestions using 0.25% Trypsin, containing most of the blood cells, were removed, and fresh digestion solution was added. Then, fresh digestion solution (0.25% dispase, 0.25% collagenase type II in HBSS; Sigma-Aldrich, Cat# H6648) (5 ml in total) was added, followed by shaking at 37°C for 40 min. The suspension was collected. The previous step of digestion and collection of suspension were repeated once more. The single-cell suspension enriched in Tdtomato+ cells (>10% Tdtomato+ by microscopy) released by the third digestion was passed through a 40 µm filter. The cells were pelleted by centrifugation and washed twice with Hanks' Balanced Salt solution (HBSS, Cat# H6648, Sigma-Aldrich), resuspended in HBSS, and the single cells encapsulated for library preparation and sequencing using the Chromium single cell platform (10X Genomics Inc) at the University of Hong Kong, Centre for PanorOmic Sciences (HKU CPOS), as per the manufacturer's protocol. Library size and concentration were determined by Qubit, quantitative PCR, and Bioanalyzer assays. Viabilities of 69 and 74.5% were recorded for the P6 and P56 samples, respectively. A raw total input of 22,750 cells was estimated for each sample.

## Single-cell RNA sequencing and bioinformatic analyses

Library preparation and Illumina (NovaSeq 6000) sequencing (151 bp paired end) were performed at HKU CPOS, at 150 Gbp throughput per sample. In both samples, 70% of bases had quality scores ≥Q30. Cells with high percentage mitochondrial RNA and ribosomal RNA were filtered out for quality control. The raw data were aligned to mouse genome (mm10), tdTomato sequence (1431 bp), and the Woodchuck Hepatitis Virus Post-transcriptional Regulatory Element (WPRE) sequence (593 bp), using Cell Ranger (v3.1.0). The R26 tdTomato cre-reporter vector contains the WPRE sequence downstream of the tdTomato sequence to reduce readthrough transcription (*Higashimoto et al., 2007*). Since the current single-cell technology generates sequences ~500 bp upstream from the 3' end of transcripts, tdTomato sequences may not always be detected due to the addition of WPRE. We confirmed that tdTomato and WPRE are linearly correlated but with enhanced detectability in the latter (*Figure 1—figure supplement 1E and F*). As such, we used WPRE as a surrogate for tdTomato. We assigned a cutoff value of WPRE >14 to match our experimental determination of the frequency of tdTomato-positive HCs (~75%) (*Yang et al., 2014a*; *Tan et al., 2020*), which also corresponds to a saddle point in the WPRE distribution (*Figure 1Figure 1—figure supplement 1*). This cutoff was used to separate the osteogenic cells into HC-derived osteoblasts and non-HC-derived osteoblasts (*Figure 1C, J and K*). To isolate endochondral cells, cells with fewer than 1600 genes or clusters expressing *Ptprc* (CD45) were excluded. Data were further processed with Seurat (v3.9.9) (*Stuart et al., 2019*). In all, 3163 (average 3382 genes per cell) and 430 (average 1702 genes per cell) endochondral cells were found in the P6 and P56, respectively. Population signatures were identified by comparing every population against all other cells, using Seurat's FindAllMarkers function, with a default Wilcoxon rank-sum test used and an FDR cutoff of <0.05, and expressed percentage point difference >20%. From these signatures, eight endochondral bone populations were identified in P6 based on the literature (*Figure 1E*), including chondrocytes, pre-hypertrophic chondrocytes, HCs, proliferating chondrocytes, immature osteoblasts, and mature osteoblasts (*Ayturk et al., 2020*; *Tan et al., 2018*; *Tikhonova et al., 2019*; *Baccin et al., 2020*; *Zhou et al., 2015*). In P56, five osteogenic clusters were identified, including subpopulations of chondrocytes

and osteoblasts (*Figure 1F*). The osteogenic cells of P6 and P56 were integrated using the canonical correlation analysis (CCA) approach (*Figure 1H*). Gene Ontology analyses were performed using GSEA (gsea-msigdb.org).

## Quantitation of cells

Fluorescence images were processed with ImageJ (developed by the National Institutes of Health, USA). For young mice at P3 or P10, the number of cells was counted manually. For mice aged 2 months, images were taken using Carl Zeiss LSM confocal 800 system (Carl Zeiss, Oberkochen, Germany). Overlap of DAPI with RFP (tdTomato), GFP, or OSTERIX was considered as one single cell as in *Figure 6*. Three separate bone sections from each animal were counted blindly in a region below the chondro-osseous junction. Cells were quantitated using ImageJ software and averaged by a specified area 4 mm below chondro-osseous junction for 2-month-old mice. Statistical analyses were performed using GraphPad Prism 8.0. Predetermined sample sizes were not used in statistical methods. Unpaired two-tailed Student's *t*-test was used. $p < 0.05$ was considered statistically significant.

For quantification of vascularized area below the chondro-osseous junction, blood vessels are stained with the endothelial cell marker ENDOMUCIN (EMCN). Vascularized area below the chondro-osseous junction is measured by manual drawing of blood vessel area circled by ECMN staining with ImageJ. Measured area is represented as $mm^2$. Quantification of osteocytes were identified by their location in cortical bone matrix.

## Microcomputed tomography (microCT) imaging

The tibia and femora were collected and fixed in 100% absolute ethanol overnight at 4°C. The tibia was sent to scanning by the eXplore Locus SP System (GE Company, UK). Acquired images were processed using the Dataviewer, CTAnalyser, and CTVolume from Bruker, USA. Morphometric values, including bone mineral density (BMD), bone volume/tissue volume (BV/TV), trabecular thickness (Tb.Th), trabecular separation (Tb.Sp), and trabecular number (Tb.N), are presented. Samples from *Mmp14*$^{F/F}$;C10Cre and *Mmp14*$^{F/-}$;C10Cre were pooled for microCT analysis.

## Histology, immunostaining, and in situ hybridization

Hematoxylin and eosin (H&E) staining and immunostaining were performed as previously described (*Yang et al., 2014a*). Briefly, mouse tibiae were fixed in 4% (weight/volume) paraformaldehyde over-night at 4°C with agitation. Tissues were decalcified in 10% EDTA for three washes for 3 days. The samples were further dehydrated and embedded in wax. Immunostaining and in situ hybridization were performed as previously described. Samples from *Mmp14*$^{-/-}$, *Mmp14*$^{+/+}$, *Mmp14*$^{F/F}$;C10Cre, *Mmp14*$^{F/-}$;C10Cre, and *Mmp14*$^{F/+}$;C10Cre were analyzed.

## Primary osteoblast culture

Primary murine osteoblasts obtained by from mice were identified by C1-GFP fluorescence in vitro. Extraction of trabecular osteoblasts was described in *Chan et al., 2012*. Tibiae and femora and the cartilage were carefully removed at P14. The entire tibia and femur were subjected to centrifugation to remove the bone marrow. Cortical bone was also removed and the remaining trabecular were subjected to serial digestion. Cells released after the third and fourth digestion were collected and the tdTomato-positive cells were clearly seen under fluorescent microscopy. The HC descendants were first cultured in alpha-minimum essential medium (αMEM) supplemented with 50 μg/ml penicillin/streptomycin, 12.5 μg/ml gentamycin, and 0.5 μg/ml fungizone for 3 days and were subsequently differentiated in αMEM containing 50 mg/ml ascorbic acid, 10 mM β-glycerophosphate, 10 μg/ml collagen type I (StemCell Technologies, Cat# 07001), 50 μg/ml penicillin/streptomycin, 12.5 μg/ml gentamycin, and 0.5 μg/ml fungizone for 7 days. Matrix-producing osteoblasts (identified by C1-GFP fluorescence) were then challenged with $1 \times 10^{-7}$ M PTH(1–34) and collected at indicated time points. Cells were washed with ice-cold PBS and lysed using Radioimmunoprecipitation assay (RIPA) buffer (150 mM NaCl, 0.5% sodium deoxycholate, 0.1% sodium dodecyl sulfate [SDS], 50 mM Tris-HCl, pH 8.0 supplemented with cOmplete, EDTA-free Protease Inhibitor Cocktail [Roche, Cat# 11873580001]). Western blotting was performed as previously described (*Chan et al., 2012*).

## Transfection and cell line experiments

Human embryonic kidney 293 (HEK293) obtained from ATCC CRL-1573. Mycoplasma screening is performed every 1–2 months and our cells are mycoplasma-free. 293 cell were verified by STR profiling. For testing in vitro cleavage of PTH1R by MMP14, HEK293 cells were seeded on 12- or 6-well dishes at 30% confluence and transfected with vectors (pcDNA3.1) expressing MT1-FL (full-length MMP14), MT1-EA (inactive MMP14), and PTH1R at 60–70% confluence with Fugene HD (Promega) according to the manufacturer's instructions. Cells were collected 48–72 hr after transfection. To test for cellular response to PTH, stable cell line expressing PTH1R was prepared by transfecting 293 cells with pcDNA3.1-PTH1R and was selected using G418 according to the manufacturer's instructions. Stable clones expressing PTH1R were examined by western blotting. To test for response to PTH, $1 \times 10^{-7}$ M PTH was added 48–72 hr after transfection with MT1-EA or MT1-FL and cell was collected at corresponding time points.

## cAMP ELISA assay

Cyclic adenosine monophosphate (cAMP) was measured using cAMP cOmplete ELISA kit from Enzo Life Sciences (Cat# ADI-900-163). 293T Cells expressing PTH1R were transfected with empty vector, Mmp14 and Mmp14-EA. After 48–72 hr, the cells were challenged with $1 \times 10^{-7}$ M PTH(1–34) and subsequently lysed in 0.1 M HCl in 0, 30, and 60 min time series. The supernatant was collected and assayed according to the manufacturer's instructions.

## Selection of PTH1R expressing stable clone

To establish a 293 cell line that stably expresses PTH1R, a pcDNA3 vector from Invitrogen containing a neomycin selection gene, a human PTH1R coding sequence with a modified HA-tag at exon2 provided by Prof. Martin J. Lohse and a ORF human PTH1R (GenScript, Cat# OHu15045D) was transfected into 293 cells. After 2 days, the transfected cells were selected with G418 at 1 mg/ml for approximately 1 week. Individual clones was picked by pipetting and screened for PTH1R expression by western blot.

## Extraction of protein from mouse trabecular bone

Trabecular bones were harvested from the $Mmp14^{-/-}$ and littermate control mice, followed by homogenizing bones in RIPA with a bullet blender for 15 min at 4°C. After leaving on ice for 10 min, the samples were centrifuged at $13,000 \times g$ for 15 min at 4°C. Next, the supernatants were collected and processed for Bio-Rad protein assay for protein concentration determination.

## In vitro cleavage of rPTH1R and peptide fragments from PTH1R

Peptide fragments covering amino acids 55–66 (GenScript) and His-tagged recombinant extracellular domain of Human PTH1R (Cat# 12232-H08H, Sino Biological) were used. The peptides and recombinant proteins were incubated with recombinant MMP14 (BML-SE259-0010, Enzo Lifesciences) for 16 hr, and the mixture was analyzed using western blotting or MALDI-TOF mass spectrometry (Centre for Proteomics, the University of Hong Kong).

## Western blot analysis

Western blotting was performed as described previously (*Chan et al., 2012*). Briefly, RIPA cell lysis buffer was prepared using 150 mM NaCl, 1% NP-40, 50 mM Tris, 0.1% SDS, and 0.1% sodium deoxycholate. Protein lysate was then blotting on a polyvinylidene fluoride (PVDF) membrane (Millipore, MA) and transferred for 1 hr at 15 V using Trans-Blot SD Semi-Dry Transfer Cell (Bio-Rad). The PVDF membrane was blocked with 5% milk in Tris-buffered saline with 0.1% tween-20 (TBST) for 1 hr and incubated with agitation with primary antibody overnight at 4°C. The membrane was further incubated in secondary antibody conjugated to HRP diluted in blocking buffer for 1 hr at room temperature with agitation. Nonspecific binding of HRP secondary antibody to the PVDF membrane was washed for three times with TBST each for 3–5 min. To develop the signals for chemiluminescence of the targeted proteins, WesternBright ECL kit (K-12045-D50) from Advansta was used according to the manufacturer's instructions. After incubation with the ECL substrate, the PVDF membrane was taken to the ChemiDocTM MP Imaging system for the visualization of target proteins.

To detect the cleaved fragments of PTH1R by western blotting, the cell lysate was either not boiled or heated to 70°C for 5 min for denaturation. Boiling will cause PTH1R containing 7-transmembrane

domain to aggregate. EDTA was added to the RIPA cell lysis buffer to a concentration of 5 mM to inhibit membrane metalloproteinase activity. PTH1R was deglycosylated by 1 hr (secondary cell lines) or overnight incubation (primary cells or tissue lysate) with PNGaseF at 37°C (NEB, Cat#, P0704S). 1 µl of 1 M DTT was added to 50 µl of cell lysate to ensure complete dissociation of disulfide bonds linking cleaved extracellular domain of PTH1R. The processed protein fragments were analyzed by SDS-PAGE.

### TUNEL labeling and Edu incorporation assay

Proliferation and apoptosis assays were described in *Yang et al., 2014a*. Edu labeling was performed with intraperitoneal injections of Edu at 150 mg/kg body weight, and the mice were sacrificed 2 hr later. Labeled cells in paraffin sections were detected using Click-iTR EdU imaging kits from Invitrogen. In situ terminal deoxynucleotidyl transferase deoxyuridin triphosphate nick end labeling (TUNEL) assay was performed using in situ cell death kit from Roche (Basel, Switzerland).

### X-gal staining

β-galactosidase staining was performed as previously described (*Yang et al., 2014a*). Briefly, harvested embryos were prefixed in 4% PFA for 1 hr at 4°C. Prefixed samples were washed in washing buffer three times and were stained in X-gal solution overnight.

### Co-immunoprecipitation assay

Co-immunoprecipitation assay was performed as previously described (*Chan et al., 2012*). Briefly, in vitro cultured cells were transfected with PTH1R-HA and MT1-FL expression plasmids. After 48–72 hr, cells lysed using ice-cold IP lysis buffer (pH 7.5, 50 mM Tris-HCl, 150 mM NaCl, 1% NP-40, 0.5% sodium deoxycholate, one tablet protein inhibitor cocktail, 5 mM EDTA) containing proteinase inhibitor cocktail. The cell lysate was centrifuged at $13,000 \times g$ for 10 min at 4°C, the supernatant was cleared with 50 µl of Protein agarose G (Roche, Cat# 11719416001) for 3 hr at 4°C and subsequently incubated with 50 µl of Protein agarose G with corresponding primary antibody overnight at 4°C. The protein agarose complex was pelleted, resuspended with IP buffer two times, followed by washing in high salt (pH 7.5 50 mM Tris-HCl, 500 mM NaCl, 0.1% NP40, 0.05% sodium deoxycholate) and low salt (pH 7.5 50 mM Tris-HCl, 0.1% NP40, 0.05% sodium deoxycholate) IP buffer. 30 µl 1× loading buffer was used to the antibody and G-agarose complex and heated to 65°C for 10 min, followed by SDS-PAGE and western blotting.

### TRAP staining

Osteoclasts were detected by tartrate-resistant phosphatase (TRAP) staining. After rehydration of sections, they are incubated with Napthol-Ether Substrate at 37°C for 1 hr. After that, they are changed directly to Pararosaniline solution freshly made by mixing 1:1 ratio of sodium nitrile solution and Pararosaniline Dye for color development. TRAP-positive, multi-nucleated (>3 nuclei) osteoclasts are stained by red color and the section is counterstained by hematoxylin for contrast.

### Statistics

Results were represented as mean ± SD. Statistical evaluation was done by nonparametric two-tailed Student's *t*-test using GraphPad Prism version 8 for Microsoft (http://www.graphpad.com). When ANOVA detected a significant interaction between the variables or a significant main effect between genotype or treatment, a Tukey's post hoc was used to determine the significance for multiple comparisons. For in vitro experiment with measurement of gel blot intensity and cAMP level, ANOVA followed by unpaired *t* Welch's correction was used to determine statistical significance. $p < 0.05$ was considered significant.

## Acknowledgements

We thank Henry Kronenberg for sharing mouse reagents and for critical input and advice on the study; Danny Chan, and Reinhard Faessler for helpful discussion, Martin Loshtse for providing the PTH1R-HA expression vector and Keith Leung for help with figure preparation. This work was supported by grants to KSEC from the Research Grants Council and University Grants Council of Hong Kong: AoE/M--04/04, T12-708/12N, the Health and Medical Research Fund (07183766) and the Jimmy and Emily

Tang Professorship. PKC (no. 20210830100C) and ZJT (no. 20210802658C) were both supported by the Shenzhen Peacock Plan.

## Additional information

### Competing interests

Kathryn Song Eng Cheah: Senior editor, eLife. The other authors declare that no competing interests exist.

### Funding

| Funder | Grant reference number | Author |
|---|---|---|
| Research Grants Council, University Grants Committee | AoE/M-04/04 | Kathryn Song Eng Cheah |
| Research Grants Council, University Grants Committee | T12-708/12N | Kathryn Song Eng Cheah |
| Health and Medical Research Fund | 07183766 | Kathryn Song Eng Cheah |
| University of Hong Kong | Jimmy & Emily Tang Professorship | Kathryn Song Eng Cheah |

The funders had no role in study design, data collection and interpretation, or the decision to submit the work for publication.

### Author contributions

Tsz Long Chu, Conceptualization, Data curation, Formal analysis, Validation, Investigation, Visualization, Methodology, Writing – original draft, Writing – review and editing; Peikai Chen, Data curation, Formal analysis, Investigation, Methodology, Writing – original draft, Writing – review and editing, Bioinformatics analyses; Anna Xiaodan Yu, Formal analysis, Validation, Investigation, Methodology, Writing – original draft; Mingpeng Kong, Formal analysis, Investigation, Methodology, Writing – review and editing; Zhijia Tan, Validation, Investigation, Writing – review and editing; Kwok Yeung Tsang, Investigation, Methodology, Writing – review and editing; Zhongjun Zhou, Resources, Supervision, Writing – review and editing; Kathryn Song Eng Cheah, Conceptualization, Resources, Formal analysis, Supervision, Funding acquisition, Writing – original draft, Project administration, Writing – review and editing, Data interpretation

### Author ORCIDs

Tsz Long Chu (ID) http://orcid.org/0000-0001-8553-6880
Peikai Chen (ID) http://orcid.org/0000-0003-1880-0893
Zhijia Tan (ID) http://orcid.org/0000-0003-2295-5169
Zhongjun Zhou (ID) http://orcid.org/0000-0001-7092-8128
Kathryn Song Eng Cheah (ID) http://orcid.org/0000-0003-0802-8799

### Ethics

Animal care and experiments were in accordance with the protocols approved by the Committee on the Use of Live Animals in Teaching and Research of the University of Hong Kong. Protocol nos: 3981-1, 5295-20, 5527-20.

### Decision letter and Author response

Decision letter https://doi.org/10.7554/eLife.82142.sa1
Author response https://doi.org/10.7554/eLife.82142.sa2

## Additional files

### Supplementary files
- MDAR checklist
- Supplementary file 1. List of antibodies and probes.

### Data availability
Sequencing data have been deposited in GEO under accession codes: GSE159544 and GSE222203. The processed data is interactively hosted on https://www.sbms.hku.hk/kclab/mmp14-hcob and part of the scripts for analyzing this data was deposited on GitHub (https://github.com/hkukclab/mmp14-hcob, copy archived at *Chen, 2022*). All data generated or analysed during this study are included in the manuscript and supporting files; source data files have been provided for Figure 1.

The following datasets were generated:

| Author(s) | Year | Dataset title | Dataset URL | Database and Identifier |
|---|---|---|---|---|
| Tan Z, Kong MP, Chen PK, Cheah KSE | 2023 | hypertrophic chondrocytes and osteogenesis in P6 mouse tibia | https://www.ncbi.nlm.nih.gov/geo/query/acc.cgi?acc=GSE159544 | NCBI Gene Expression Omnibus, GSE159544 |
| Ax Yu, Chen PK, Cheah KSE | 2023 | MMP14 cleaves PTH1R in the chondrocyte derived osteoblast lineage, curbing signaling intensity for proper bone anabolism | https://www.ncbi.nlm.nih.gov/geo/query/acc.cgi?acc=GSE222203 | NCBI Gene Expression Omnibus, GSE222203 |

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
