## [Editor Report]

The authors present novel findings that PTH signaling plays a significant role in bone formation in hypertrophic chondrocyte (HC)-derived osteoblasts and MMP14 cleaves PTH1R and inhibits PTH signaling. These studies significantly contribute to our understanding of molecular mechanisms of postnatal bone formation and adult bone remodeling, especially the HC cells in this process. The study was well-designed and well-conducted. The data in this study are convincing and support the conclusion made by the authors.

---

## [Decision Letter]

**Decision letter after peer review:**

Thank you for submitting your article "MMP14 cleaves PTH1R in the chondrocyte derived osteoblast lineage, curbing signaling intensity for proper bone anabolism" for consideration by *eLife*. Your article has been reviewed by 3 peer reviewers, including Di Chen as Reviewing Editor and Reviewer #1, and the evaluation has been overseen by Mone Zaidi as the Senior Editor. The following individuals involved in the review of your submission have agreed to reveal their identity: Mei Wan (Reviewer #2); Marc N Wein (Reviewer #3).

Essential revisions:

1) It is unclear how specific the MMP14/PTH1R pathway is in the transition of HCs to osteoblasts. Are other MMPs besides MMP14 involved in the PTH1R cleavage? Is PTH1R the only substrate of MMP14?

2) It seems that MMP14 cleaves the PTH1R after amino acid 61, which thus should generate a short fragment. How can this be reconciled with the in vivo cleavage pattern that is observed?

3) The authors need to demonstrate the co-expression of PTH1R and MMP14 in the same cells (HC-derived osteoblasts).

*Reviewer #2 (Recommendations for the authors):*

1. Figure 3E bottom panel, it would be better if the authors can show the uncropped gel image of the PTH1R blot as did in Figure 3C such that both truncated and intact PTH1R bands can be shown in one image.

2. It is important that the authors not only detected MMP14-induced PTH1R cleavage in an in vitro 293T cell culture system but also detected the truncated 55kDa form of PTH1R in trabecular bone tissue lysates (Figure 3E) and trabecular osteoblast culture (Supplementary Figure 4D). I suggest this piece of data (Supplementary Figure 4D) move to the main figures section.

*Reviewer #3 (Recommendations for the authors):*

I will not duplicate assessments about the overall impact in this section. Specific suggestions and questions for the authors are as follows:

1. For Figure 1, how confident are the authors that their digestion protocol liberates a sizeable fraction of bone-associated osteoblasts/osteocytes from the matrix? This information may be important in interpreting whether different subsets of osteoblasts are HC- or non-HC-derived. For example, the authors could examine bone fragments before and after digestion for expression of osteoblast and osteocyte marker genes. It is clear (and not surprising) that digestion efficiency varies considerably between P6 and P56 mice.

2. For Figure 1, a feature plot demonstrating tdTomato mRNA expression across different subsets would be helpful. Overall, more information is needed, at the single cell level, about which cells in these digested bone fragments do and do not express tdTomato.

3. It is confusing why the authors need to stain sections with an RFP antibody rather than examine intrinsic tdTomato fluorescence. Clarification should be provided as to why this non-standard tdTomato detection method was required.

4. For Figure 3A, the violin plots shown really do not demonstrate the co-expression of Pth1r and Mmp14 in the same cells. These plots demonstrate that groups of HC-derived osteoblasts and groups of non-HC-derived osteoblasts express widely varying levels of these two transcripts. Co-expression could be demonstrated using feature plots or other methods to reanalyze scRNA-seq data at the single-cell level.

5. Figure 3E shows an important result: that a truncated Pth1r immunoreactive band is absent in Mmp14-/- samples. The entire immunoblot here in the main figure (rather than the data supplement) should be shown here rather than cropped portions for full transparency.

6. The MMP14-induced PTH1R cleavage pattern seen in vitro is difficult to reconcile with the immunoblotting results in vivo. It seems that MMP14 cleaves the receptor after amino acid 61, which thus should generate a short fragment. How can this be reconciled with the in vivo cleavage pattern that is observed? This is an important point that should be addressed.

7. Overall, the quantified effects of MMP14 expression on biochemical PTH-stimulated endpoints are somewhat modest. The graphs in Figure 4B are difficult to follow (significance bars seem to compare EV and MT1EA) and should be clarified. Moreover, it is difficult to understand the physiologic significance of the changes in pCREB seen in MMP14-/- cells treated with PTH. Ideally, the authors might complement these biochemical studies with physiologically-important gene expression changes such as PTH-induced RANKL induction.

8. In Figure 4E, the authors should explain the meaning of the orange 'mutation sites' which don't seem to be found anywhere in the manuscript text.

9. Overall, the methods section could be reorganized to match the order of techniques as they appear in the Results section.

10. The findings in Figure 7 are interesting, but say nothing about the MMP14/PTH1R relationship studied in the rest of the manuscript. At the very least, it would make sense to look at MMP14 and PTH1R expression in situ in tdTomato-positive cells observed in the vehicle and PTH-treated aged mice.

11. The discussion could be modified to clearly note differences between cells labeled by collagen X-cre and hypertrophic chondrocytes. Most investigators in this field would agree that collagen X-creER can initially label cells in the primary spongiosa and elsewhere. So it is difficult to conclude with absolute certainty that tdTomato-positive cells in this study are all derived from hypertrophic chondrocytes.

12. Reference 55 is interesting, but there are many other studies demonstrating that intermittent PTH treatment in humans with osteoporosis does affect cortical bone (PMID 23044908 and 26964731 for example).

13. The discussion should be modified to include the possibility that substrates of MMP14 other than the PTH receptor might contribute to the bone phenotypes observed. An important priority for future study should be to generate PTH receptor mutants that cannot be cleaved by MMP14 for in vivo analysis.

---

## [Author Response]

Reviewer #2 (Recommendations for the authors):1. Figure 3E bottom panel, it would be better if the authors can show the uncropped gel image of the PTH1R blot as did in Figure 3C such that both truncated and intact PTH1R bands can be shown in one image.

Thank you for your suggestion and comments, we have revised Figure 3E which now shows the uncropped gel image.

2. It is important that the authors not only detected MMP14-induced PTH1R cleavage in an in vitro 293T cell culture system but also detected the truncated 55kDa form of PTH1R in trabecular bone tissue lysates (Figure 3E) and trabecular osteoblast culture (Supplementary Figure 4D). I suggest this piece of data (Supplementary Figure 4D) move to the main figures section.

Thank you for your comment, the figure is now moved to Main figure 3E.

Reviewer #3 (Recommendations for the authors):I will not duplicate assessments about the overall impact in this section. Specific suggestions and questions for the authors are as follows:1. For Figure 1, how confident are the authors that their digestion protocol liberates a sizeable fraction of bone-associated osteoblasts/osteocytes from the matrix? This information may be important in interpreting whether different subsets of osteoblasts are HC- or non-HC-derived. For example, the authors could examine bone fragments before and after digestion for expression of osteoblast and osteocyte marker genes. It is clear (and not surprising) that digestion efficiency varies considerably between P6 and P56 mice.

Thank you for your comments, and for recognising the challenges in obtaining single cells from hard tissues such as bone (especially more mature bone), while at the same time retaining good quality mRNA. For that reason we used sequential protease digestions and the cells encapsulated for sequencing after the third digestion (see Methods section). We do recognise that we may not have isolated all the cells from the trabecular bone. The main purpose of the experiments were to identify candidate protease mediators of the HC transition to osteoblasts by isolating and analysing HC descendants. We were not aiming to make a comprehensive quantification of all populations in the trabecular bone. However, given that we identified osteogenic cells of differing maturity (pre-osteoblasts, mature osteoblasts and osteocytes) as identified by known molecular signatures, this suggests we have a good representation of the different HC-derived and non-HC-derived populations. Our identification of HC-derived cells is based on the detection of cells expressing the lineage marker, tdTomato and its surrogate sequence, WPRE (see Methods), and osteogenic cells which do not express tdTomato/WPRE. The proportion of TdTomato positive osteogenic cells identified is in broad agreement with the frequency determined from the quantitation of HC-derived cells in tibia sections (approximately 30% at P6) as shown in Figure 1H and Figure 1 —figure supplement 1. Of note, we found 25% of *Sost* -expressing cells are tdtomato positive, suggesting the method of extraction had succeeded in releasing the most mature osteogenic cells. The broad agreement between the frequencies of the single cell populations and the quantitation of HCderived cells in equivalent stage tibia, suggest that the data presented is representative of the in vivo state. Please note that in the revised manuscript we have substituted new P56 scRNAseq data derived from *MMp14 +/-* tibia (mice have normal phenotype) as we were able to isolate more cells and we think this is a more representative dataset.

2. For Figure 1, a feature plot demonstrating tdTomato mRNA expression across different subsets would be helpful. Overall, more information is needed, at the single cell level, about which cells in these digested bone fragments do and do not express tdTomato.

The purpose of the single cell transcriptome assays was to identify candidate facilitators of the HC to osteoblast transition, which could then be followed up functionally. In Figure 1 —figure supplement 1., we showed that 18%, 33% and 25% percent of immature Obs, mature Obs and osteocytes respectively are tdtomato positive respectively. Overall the tdtomato mRNA expression comprises 20-35% in different subset. We have added Figure 1M which shows the frequency of co-expression of *Mmp14* and *Pth1R* in HC-derived and non HC derived cells. We also have added Figure 3—figure supplement 1D showing coexpression of *Pth1r, Mmp14, WPRE* and *GFP (*representing the *Col1a1* reporter) at P56, which suggest that the majority of HC-derived osteogenic cells co-express *Pth1r* and *Mmp14*.

3. It is confusing why the authors need to stain sections with an RFP antibody rather than examine intrinsic tdTomato fluorescence. Clarification should be provided as to why this non-standard tdTomato detection method was required.

Thank you for your comment, the cat. No. of the antibody we used is AB1140100. http://www.sicgen.pt/product/rfp-polyclonal-antibody_1_181. It also reacts with tdtomato and we found that this antibody gave stronger signals.

4. For Figure 3A, the violin plots shown really do not demonstrate the co-expression of Pth1r and Mmp14 in the same cells. These plots demonstrate that groups of HC-derived osteoblasts and groups of non-HC-derived osteoblasts express widely varying levels of these two transcripts. Co-expression could be demonstrated using feature plots or other methods to reanalyze scRNA-seq data at the single-cell level.

Thank you for your comment, please see answer to comment 2 above.

5. Figure 3E shows an important result: that a truncated Pth1r immunoreactive band is absent in Mmp14-/- samples. The entire immunoblot here in the main figure (rather than the data supplement) should be shown here rather than cropped portions for full transparency.

Thank you for your comment, in the revised Figure 3E the original uncropped western blot is shown.

6. The MMP14-induced PTH1R cleavage pattern seen in vitro is difficult to reconcile with the immunoblotting results in vivo. It seems that MMP14 cleaves the receptor after amino acid 61, which thus should generate a short fragment. How can this be reconciled with the in vivo cleavage pattern that is observed? This is an important point that should be addressed.

This is a very good question. For cell line experiments, the truncated PTH1R appears at around 55kDa in HEK293 cells transfected with PTH1R and in vivo Mmp14+/+ trabecular bone lysate, suggesting a proteolytic event is possible at around amino acid ~90. To further investigate this proteolytic event, a recombinant extracellular human PTH1R-ECD (with amino acid 1-181) labeled with polyhistidine at the C-terminus from Sinobiologicals was incubated with recombinant rMMP14 (Figure 3D). After deglycosylation, the uncleaved full length PTH1R-ECD appears at ~25kDa (Figure 3D lane3) in SDS-PAGE with a theoretical predicted molecular size of 21kDa (a 4kDa difference from theoretical). Cleavage of PTH1RECD by rMMP14 results in one peptide observed at ~20kDa and another peptide observed at ~15kDa (lane2). These results indicate multiple cleavage of PTH1R-ECD by rMMP14 is likely, with one cleavage site at around amino acid 61, and another cleavage site at amino acid 90~100. To confirm one of the exact cleavage sites of PTH1R by MMP14, a peptide from amino acid 55~65 was synthesized (Genscript) and incubated with rMMP14. The resulting mixture was analyzed by mass spectrometry and the data confirmed amino acid 61 is one of the MMP14-PTH1R cleavage sites. In summary the rPTH1R fragments involved in in vitro experiments are,

1. Deglycosylated uncleaved PTH1R-ECD + histag theoretical 21kDa, observed 25kDa (Figure 3D lane 1 and 3)

2. Truncated PTH1R-ECD from 62 to 181 + histag theoretical 15kDa, observed 20kDa (lane 2 top band)

3. Truncated PTH1R-ECD from around ~100 to 181 + histag theoretical 10kDa, observed 15kDa (lane 2 second band)

4. Cleaved PTH1R-ECD from 1 to 61, no histag, not detectable in SDS-PAGE by his-tag antibody

Therefore, the recombinant protein cleavage experiment using extracellular PTH1R-ECD is consistent with cell line and in vivo results with truncated PTH1R at ~55kDa, given that the theoretical size of truncated PTH1R, from amino acid 62 to 593 at 59kDa and from amino acid 101 to 593 at 55kDa. We have added a detailed explanation to the manuscript under the Results section titled ”PTH1R is a substrate of MMP14”.

7. Overall, the quantified effects of MMP14 expression on biochemical PTH-stimulated endpoints are somewhat modest. The graphs in Figure 4B are difficult to follow (significance bars seem to compare EV and MT1EA) and should be clarified. Moreover, it is difficult to understand the physiologic significance of the changes in pCREB seen in MMP14-/- cells treated with PTH. Ideally, the authors might complement these biochemical studies with physiologically-important gene expression changes such as PTH-induced RANKL induction.

Thank you for your question. It has been widely accepted that p-CREB is an immediate downstream readout of PTH signalling (2, 3). It is therefore relevant to test the Mmp14-PTH signaling axis by measuring the effect on pCREB levels (Figure 4A). We have added to Figure 6—figure supplement 1C images showing increased Cyclin D and Runx2 expression in *Mmp14ΔHC* mutants, suggesting increased downstream action of PTHMMP14 axis which suggests a direct effect on osteogenesis and cell cycle regulation in HCderived cell fraction.

8. In Figure 4E, the authors should explain the meaning of the orange 'mutation sites' which don't seem to be found anywhere in the manuscript text.

Thank you for your comments. The diagram has been corrected in the revised version.

9. Overall, the methods section could be reorganized to match the order of techniques as they appear in the Results section.

Thank you for your comment. The methods section has been rearranged to match the results.

10. The findings in Figure 7 are interesting, but say nothing about the MMP14/PTH1R relationship studied in the rest of the manuscript. At the very least, it would make sense to look at MMP14 and PTH1R expression in situ in tdTomato-positive cells observed in the vehicle and PTH-treated aged mice.

Thank you for your comment, we have added to Figure 7—figure supplement 1 showing induction of MMP14 after PTH treatment in 1 year old mice, suggesting our finding is applicable in adults.

11 The discussion could be modified to clearly note differences between cells labeled by collagen X-cre and hypertrophic chondrocytes. Most investigators in this field would agree that collagen X-creER can initially label cells in the primary spongiosa and elsewhere. So it is difficult to conclude with absolute certainty that tdTomato-positive cells in this study are all derived from hypertrophic chondrocytes.

Thank you for your comments. That HCs can contribute to osteogenic lineage has been confirmed by many studies. It has been shown in a pulse and chase lineage tracing experiment (4) that at 16h Col10a1-CreERT initially labels predominantly HCs and very few cells at the chondro-osseus junction but by 24h there are fewer HC labelled cells in but labelled cells are now in the primary spongiosa. This finding has been reproduced in a recent (5) study in which a 12h, 24h and 2 week pulse and chase lineage tracing experiment was performed. The specificity of *Col10a1* expression in HCs has also been well validated and it is therefore used to identify the cell of origin in lineage studies. Perhaps the reviewer was referring to *Col2a1* CreERT lineage tracing studies where indeed a broader spectrum of cells are labelled initially due to the activity in the perichondrium (see supplementary figure in (4)) and therefore HCs are not the only cell of origin when *Col2a1* CreERT is used. Of course it is also interesting that *Col10a1-cre* lineage studies show that HC s can also contribute to adipogenic and other progenitors e.g (5, 6). This was the rationale to test whether MMP14 deficiency can affect the proportion of HC-derived adipocytes.

12. Reference 55 is interesting, but there are many other studies demonstrating that intermittent PTH treatment in humans with osteoporosis does affect cortical bone (PMID 23044908 and 26964731 for example).

Thank you for your suggestion, we have added those references to the revised manuscript.

13. The discussion should be modified to include the possibility that substrates of MMP14 other than the PTH receptor might contribute to the bone phenotypes observed. An important priority for future study should be to generate PTH receptor mutants that cannot be cleaved by MMP14 for in vivo analysis.

Thank you for your excellent advice. We agree that future studies should define functionally the critical cleavage sites in PTH1R and the contribution of uncleaved PTH1R to the bone phenotype. We also agree that consideration should be given to the possibility of contribution of other MMP14 substrates. In Figure3-supplement 1 we tested various A Disintegrin And Metalloproteinase (ADAMs) for their cleavage of PTH1R. ADAMs were known to cleave various transmembrane proteins such as RANKL. We did not find that ADAM10, 15, 17 could cleave PTH1R. We have added these points to the discussion.

1. Delgado-Calle J, Hancock B, Likine EF, Sato AY, McAndrews K, Sanudo C, et al. MMP14 is a novel target of PTH signaling in osteocytes that controls resorption by regulating soluble RANKL production. Faseb J. 2018;32(5):2878-90.

2. Xiong L, Xia WF, Tang FL, Pan JX, Mei L, Xiong WC. Retromer in Osteoblasts Interacts With Protein Phosphatase 1 Regulator Subunit 14C, Terminates Parathyroid Hormone's Signaling, and Promotes Its Catabolic Response. EBioMedicine. 2016;9:45-60.

3. Qiu T, Wu X, Zhang F, Clemens TL, Wan M, Cao X. TGF-β type II receptor phosphorylates PTH receptor to integrate bone remodelling signalling. Nat Cell Biol. 2010;12(3):224-34.

4. Yang L, Tsang KY, Tang HC, Chan D, Cheah KS. Hypertrophic chondrocytes can become osteoblasts and osteocytes in endochondral bone formation. Proceedings of the National Academy of Sciences of the United States of America. 2014;111(33):12097-102.

5. Long JT, Leinroth A, Liao Y, Ren Y, Mirando AJ, Nguyen T, et al. Hypertrophic chondrocytes serve as a reservoir for marrow-associated skeletal stem and progenitor cells, osteoblasts, and adipocytes during skeletal development. *eLife*. 2022;11.

6. Tan Z, Kong M, Wen S, Tsang KY, Niu B, Hartmann C, et al. IRX3 and IRX5 Inhibit Adipogenic Differentiation of Hypertrophic Chondrocytes and Promote Osteogenesis. J Bone Miner Res. 2020.